# A Brief History of Stereotactic Atlases: Their Evolution and Importance in Stereotactic Neurosurgery

**DOI:** 10.3390/brainsci13050830

**Published:** 2023-05-21

**Authors:** Alfredo Conti, Nicola Maria Gambadauro, Paolo Mantovani, Canio Pietro Picciano, Vittoria Rosetti, Marcello Magnani, Sebastiano Lucerna, Constantin Tuleasca, Pietro Cortelli, Giulia Giannini

**Affiliations:** 1IRCCS Istituto delle Scienze Neurologiche di Bologna, Bologna, Via Altura 3, 40123 Bologna, Italy; paolo.mantovani@ausl.bologna.it (P.M.); caniopietro.picciano@studio.unibo.it (C.P.P.); vittoria.rosetti@studio.unibo.it (V.R.); marcello.magnani3@studio.unibo.it (M.M.); pietro.cortelli@unibo.it (P.C.); giannini.giulia3@gmail.com (G.G.); 2Dipartimento di Biomorfologia e. Scienze Neuromotorie (DIBINEM), Alma Mater Studiorum Università di Bologna, Via Altura 3, 40123 Bologna, Italy; 3Stroke Unit- Barking, Havering and Redbrige University Hospitals NHS Trust, Queen’s Hospital, Rom Valley Way, London RM7 0AG, UK; n.gambadauro@nhs.net; 4Department of Neurosurgery, AOU “G. Martino”, Via Consolare Valeria 1, 98125 Messina, Italy; sebastiano.lucerna@tin.it; 5Neurosurgery Service and Gamma Knife Center, Lausanne University Hospital (CHUV), Rue du Bugnon 46, 1011 Lausanne, Switzerland; constantin.tuleasca@chuv.ch; 6Faculty of Biology and Medicine (FBM), University of Lausanne (UNIL), Rue du Bugnon 21 CH-1011, 1015 Lausanne, Switzerland; 7Ecole Polytechnique Fédérale de Lausanne (EPFL, LTS-5), Rte Cantonale, 1015 Lausanne, Switzerland

**Keywords:** stereotactic atlas, stereotaxis, deep brain stimulation, basal ganglia, neuroanatomy

## Abstract

Following the recent acquisition of unprecedented anatomical details through state-of-the-art neuroimaging, stereotactic procedures such as microelectrode recording (MER) or deep brain stimulation (DBS) can now rely on direct and accurately individualized topographic targeting. Nevertheless, both modern brain atlases derived from appropriate histological techniques involving post-mortem studies of human brain tissue and the methods based on neuroimaging and functional information represent a valuable tool to avoid targeting errors due to imaging artifacts or insufficient anatomical details. Hence, they have thus far been considered a reference guide for functional neurosurgical procedures by neuroscientists and neurosurgeons. In fact, brain atlases, ranging from the ones based on histology and histochemistry to the probabilistic ones grounded on data derived from large clinical databases, are the result of a long and inspiring journey made possible thanks to genial intuitions of great minds in the field of neurosurgery and to the technical advancement of neuroimaging and computational science. The aim of this text is to review the principal characteristics highlighting the milestones of their evolution.

## 1. Introduction

In the history of neurosurgery, the challenging localization of deep-sited brain regions and hidden brain areas with no direct surgical exposure has driven to the development of a surgical technique labelled ‘stereotactic’, a compound term derived from two ancient Greek words, ‘stereós’ and ‘taxis’, respectively, meaning ‘three-dimensional’ and ‘position’. Stereotaxy is, indeed, the technique which has made it possible to localize and indicate a given region of the human brain through spatial coordinates. Nowadays, previously unpredictable anatomical details can be obtained through neuroimaging stereotactic techniques; hence, direct anatomical targeting is commonly used for stereotactic procedures such as microelectrode recording (MER) and/or deep brain stimulation (DBS). Nevertheless, atlas-based information still remains an indispensable complement to high-field MR to preoperative planning. Preoperatively, the stereotactic atlas is an important reference guide to avoid targeting errors due to insufficient anatomical details and, in general, to provide basic orientation during functional neurosurgery procedures [1]. Intraoperatively, it can also function as a navigation tool and provide information about the neuroanatomy surrounding the target area, the structures along the stereotactic path, and the spatial relationships between the electrode tip and critical structures such as the optic tract [2]. Additionally, it can aid in data storage. Postoperatively, the atlas can be used to evaluate the spatial positioning of deep brain stimulation (DBS) or lesion placement [3].

For such reasons, both brain atlases derived from appropriate histological techniques applied on post-mortem human brain tissue and those based on neuroimaging and functional information continue to represent an indispensable tool for both neurosurgeons and neuroscientists. Here, we review momentous advances in the history of brain atlases highlighting their principal features.

## 2. The Discovery of Brain Localization and Cranio-Cerebral Topography before Atlases

The first experimental evidence of brain localization was presented by Broca and Hughlings Jackson. In fact, already in 1861, Paul Broca defined the brain area responsible for the articulation of speech, following a meticulous study of clinical and postmortem findings of a series of patients with left hemisphere brain lesions [4]. Only three years later, in 1863, Hughlings Jackson’s description of the primary motor area was the remarkable outcome of a series of observations performed on epileptic patients [5]. Broca’s and Jackson’ discoveries were later confirmed through a series of experimental studies conducted first on animals by Fritsch and Hitzig; Ferrier; Grünbaum; Sherrington; and then on humans through the pioneering techniques of intraoperative stimulations adopted by Horsley and Cushing [6,7]; thus, the foundation stone of modern experimental neuroscience was laid. Within this context, Broca’s introduction of cranio-cerebral topography represented a fundamental contribution to the evolution of modern neuroscience. Broca aimed at providing surgeons with anatomical coordinates that would allow precise operative exposure and the recognition of the eloquent cortical areas. Therefore, he described the main cranio-cerebral correlations, including those he had previously adopted in the procedure of Broca–Championnière in order to localize the center of expressive language [8]. These cranio-cerebral correlations enabled him to accomplish the first craniotomy based on cerebral localization. Broca supplemented the previous work of Gratiolet with new contributions pertinent to the anatomy of brain sulci, fissures, and gyri connections. Cranio-cerebral topography, which was further developed thanks to the pioneering contribution of Turner and, later, Reid in England; Poirier and Chipault in France; and Taylor and Haughton in Ireland, among others, can now be rightly considered the precursor of stereotactically guided neurosurgery [9,10,11,12,13]. We owe to Robert Henry Clarke and Sir Victor Horsley the most accurate presentation of the principles of stereotaxis as well as its related device [14]. In fact, they designed and developed an ‘apparatus’, later known as the Horsley–Clarke Device, in order to study the cerebellar function in the monkey; it was based on the reproducibility of the relationships between skull’s landmarks (external auditory canals, inferior orbital rims, midline) and anatomical structures of the experimental animal’s brain. The cranial fixation points established the baselines of a three-dimensional Cartesian stereotactic coordinate system. A similar system, based on cranial landmarks, is currently used for experimental studies on rats’ brains and represents the most commonly adopted atlas by Paxinos [15].

## 3. The Invention of the Stereotactic Atlas

In 1952, Prof. Spiegel and his pupil Dr. Wycis, following their application of the Horsley–Clarke apparatus to the human stereotactic neurosurgery (Figure 1), realized that this surgical pattern required more precise planning based on brain landmarks instead of the cranial landmarks used until then [16]. In addition, in order to allow the identification of such landmarks, they used fixed post-mortem brain specimens, prior to opening the skull. Hence, throughout a systematic search for reliable, radiographically demonstrable, reference points on which to base stereotactic atlas planes of section and surgical procedures, they used a stereotactic frame to the specimens and passed metal rods via the skull and cerebrum, at known distances from each other, and at known stereotactic reference points in one or more planes. The pineal gland calcification, visible on plain x-ray films, was used as the initial reference point, leading to the definition of the intracranial space. Yet, their later decision to abandon it was due to the extreme spatial variability detected of 12 mm or more in the anteroposterior axis, and up to 16 mm in the interaural axis, which was incompatible for such precise procedures. The ensuing intuition would influence the future of stereotactic neurosurgery. Thanks to lumbar pneumography, the visualization of the posterior commissure (PC) as well as the foramen of Monro (FM) and, in some instances, of the anterior commissure (AC) was possible. The two scientists then, adopting a line connecting the center of the PC with the pontomedullary sulcus, also commonly visible at the posterior border of the pons (PO), defined an imaginary baseline, namely, the CP-PO line, in order to create their first atlas [16]. By 1962, the year of publication of Stereoencephalotomy (Part II) by E. A. Spiegel and H. T. Wycis, the standard stereotactic reference system was already based on the anterior commissure line and the posterior commissure line (AC-PC line) or intercommissural line (IC line) due to the work of Jean Talairach, a visionary French psychiatrist and neurosurgeon [17].

## 4. The Proportional System of Talairach

Following Jean Talairach’s introduction of the use of combined positive-contrast and air ventriculography, the AC and PC could be reliably shown. He also invented a relocatable stereotactic instrument that utilized teleradiographic techniques and a ‘double grid’ for a stereotactic localization system as well as integrated angiography and ventriculography in order to create, probably, the most advanced stereotactic system in the pre-CT era [18]. Talairach showed that the deep nuclei of interest in functional procedures generally maintained a consistent relationship with the intercommissural line (the line between the AC and PC) and its derivative planes (midsagittal plane, horizontal intercommissural plane, and the two vertical planes passing through the AC (VCA) and PC (VCP), respectively). Subsequently, the two vertical planes were abandoned in favor of a single intercommissural plane. The intercommissural line can be easily identified on ventriculography (Figure 2), with the contextual advantage of delimiting the thalamic and subthalamic territories.

Indeed, this line passes just above the subthalamic body of Luys and divides the central median from the red nucleus. Talairach’s most important intuition was, probably, the method to proportionately subdivide the geometric forms outlined by the IC line and the roof of the thalamus seen on lateral ventriculogram. Hence, the adoption of his proportional system, which excluded the use of absolute measures (e.g., millimeters) made it possible to record a given structure on a surface defined by points. According to Talairach’s system, the adaptation along the antero-posterior dimension is based upon two deep brain ventricular landmarks (AC and PC) (Figure 2), whereas the adaptation along the medio-lateral and cranio-caudal axes depends on the overall size of the cerebral cortex. As a consequence, neurosurgeons, by drawing Talairach’s diagram directly on the ventriculograms of the patient (Figure 3), would be enabled to reconstruct a properly scaled atlas template, from which to derive stereotactic coordinates.

## 5. The ‘Schaltenbrand and Bailey’ Atlas

In 1959, Schaltenbrand and Bailey published a brain atlas whose coordinate system seems to derive from Talairach’s space although it shows slight differences [19]. As a matter of fact, the Talairach method allows for proportional measurement of the relative distances of the various nuclei from standard reference points by using a double grid system on the single patient: the localization is more tailored to the single patient, but it requires more invasive imaging techniques. On the other hand, the Schaltenbrand atlas is more essential but reports the distances in a more rigid way, based on the measurements provided on microscope sections and without the proportional system verification. The frontal sections are displayed four per page at 4× magnification, with a scaled and labelled transparent overlay attached to each page. The 16 sections, each with the thickness of 1–4 mm and all cut from the same brain, span the region from 16.5 mm anterior to 16.5 mm posterior to the midcommissural plane. The sagittal series is presented in the same manner, but the sections on each page are one or two. The 18 sections are cut at 0.5–2.5 mm intervals, spanning the region between 2.0 and 27.5 mm lateral to the midline. Schaltenbrand and Bailey’s myelin-stained sagittal series were widely used because the majority of functional stereotactic operations involve a transfrontal (precoronal) approach to the thalamus or upper midbrain through a parasagittal entry point. The horizontal series, such as the frontal one, is presented at four planes per page at 4× magnification. The 20 sections, all cut from a single brain, span the region from 16 mm above to 9.5 mm below the midcommissural point.

## 6. An Atlas Showing Variations in Human Diencephalon

Volume 2 of the ‘Variations and Connections of the Human Thalamus’ called ‘Variations of the Human Diencephalon’, published in 1972 by Van Buren and Borke, is one of the four main brain atlases of the pre-CT era [20]. The atlas shows the variations in the topographic relationship of thalamic nuclei and basal ganglia. Thalamic nuclei and their stereotactic coordinates are shown in connection with the IC line and the sagittal plane. Each series of cresyl-violet-stained plates is reproduced one per page, at 8–10× magnification. The set of sagittal images comprises ten slices at 10× magnification, covering the area ranging from 2 to 25 mm lateral to the midline at intervals of 0.5–4 mm. These plates feature clear delineations of nuclear groups and tracts, accompanied by anatomical labels, coordinate index marks, and a magnification scale. The horizontal series, on the other hand, is made up of eight sections cut with high precision parallel to the intercommissural plane and presented at 8× magnification. The region spanning from 17 mm above to 8.1 mm below the IC line is covered by sections that are approximately 3.5 mm thick. The ten transverse (frontal) sections are photographed at 10× magnification, which is the same as the are series. The photographic plates depict a distance of 28.1 mm, ranging from 23.4 mm anterior to PC to 4.7 mm posterior to PC. The last chapter of the atlas provides comparable data for the gross anatomic structural outlines of 25 hemispheres that are normalized to either the AC or the PC. Additionally, simplified diagrams displaying the region of densest overlap (median values) and extreme ranges are included.

According to the atlas, midline ganglia have only slight variations in the horizontal and coronal planes, but the variation in the laterally located structures in the sagittal plane introduces a disturbing factor that renders quite inaccurate any system of coordinates. The aim of the atlas is to aid the surgeon to choose the set of coordinates which can give the best chances of reaching a given nucleus.

## 7. The ‘Talairach’ and ‘Schaltenbrand’ Atlases of the CT Era

The Schaltenbrand–Wahren brain atlas was published in 1977 [21]. It differs from the Schaltenbrand and Bailey atlas, being based on myelin-stained sections instead of histological sections. The atlas contains 100 instead of 97 sections and it is overall organized in a different manner, with different levels of detail and different areas covered. The Schaltenbrand–Wahren atlas includes 34 macro-series photographs, all at 2× magnification and divided into three series as follows: 19 frontal planes from 57 mm anterior to 44 mm posterior to AC (Figure 4A), five sagittal planes from the midline (0 plane) to 22 mm lateral to the midline, six horizontal planes from 18 mm above to 20 mm below the IC line from one brain, and four additional horizontal planes from 5 mm to 28 mm below the intercommissural line from another brain. The authors recognized the importance that axial imaging would play in the future and gave relevance to the horizontal unstained macro-series and the myelin-stained micro-series. All horizontal sections are parallel to the intercommissural plane. The myelin-stained brainstem series comprises 21 planes in the transverse direction, as discussed earlier, and is not the only representation available. In fact, the atlas also includes the three conventional planes, resulting in a total of 78 myelin-stained atlas photographs. The 20 frontal planes extend from 16.5 mm anterior to 16.5 mm posterior to the midcommissural plane, 17 sagittal planes span from 1.5 to 27.5 mm lateral to the midline, and 20 horizontal planes cover a distance of 16 mm above to 9.5 mm below the IC line (as shown in Figure 4B).

The stereotactic atlas by Talairach and Tournoux was published in 1988 [22]. Unlike the other atlases above mentioned, this atlas is not focused on basal nuclei, but it is more dedicated to some modern stereotactic procedures including biopsies and radiosurgery.

The Talairach proportional grid system, which is presented in three dimensions, incorporates orthogonal reference planes based on the midline, the intercommissural plane, and two ‘verticofrontal’ planes intersecting the anterior and posterior commissures, similar to Talairach’s earlier atlases. The authors note that direct distances between points in the brain can vary greatly between individuals, particularly the further from the IC line a point of interest is located. Thus, Talairach and Tournoux divide the brain into cuboidal and rectangular prism-shaped parcels called ‘orthogonal parallelograms’ (Figure 5). Each hemisphere is subdivided into nine major parcels in length (A–I along the IC line), four parcels wide (a–d along the transverse plane orthogonal to the midline and IC plane), and twelve parcels high (1–12 in vertical planes parallel to those defined by the commissures). The size of each parcel is determined based on 1/8 of the distance between the IC line and the highest point of the parietal cortex and 1/4 of the distance between the IC line and the lowest point of the temporal cortex for height (parcels 1–12), 1/4 of the distance from AC to the frontal pole, 1/4 of the distance from PC to the occipital pole, the entire distance (subdivided into thirds) between AC and PC for length (A–I), and one-fourth of the distance from the midline to the most lateral point of the parietotemporal cortex for width (4 parcels per hemisphere). Therefore, each voxel represents a fixed proportion rather than a rigid distance within the brain.

### 7.1. Consistencies and Inconsistencies of the Talairach and Tournoux 1988 and the Schaltenbrand and Wahren Atlases

Due to the methodology of their construction, traditional printed atlases have potential limitations due to 3D inconsistencies and spatial sparseness. For instance, the Schaltenbrand axial plates were not acquired exactly in the intercommissural plane but are rotated 7 degrees clockwise [23]. Additionally, the atlases are constructed based on a few brains only: the Talairach and Tournoux atlas on a single brain and the SW atlas microseries on two different brains (three various hemispheres) despite using 111 brains as the initial material. Niemann and Nieuwenhofen analyzed the 3D positions of 21 anatomical structures in the third series of the Schaltenbrand and Wahren atlas, after digital interpolation and volumetric representation [24]. Three-dimensional rendering showed that the thalamus is 10% larger than the frontally represented and 40% larger than the horizontally sectioned thalamus, sagittally. Thus, in order to match it to the sagittal series, the frontal series has to be widened in lateral direction by 19%, and it has to be compressed by 5 and 9%, respectively, in the anteroposterior and dorsobasal (vertical) direction. In contrast, the distance of structures from the midline in the horizontal and sagittal series is very similar. The horizontal series is, however, 25% smaller than the sagittal anteroposterior series and 17% in the vertical direction. On average, thalamic nuclei in the right hemisphere of brain LXXVIII (horizontal microscopic series). Spatial overlap between corresponding thalamic nuclei from the three series amounted to only 0 ± 28% when superimposed in the AC-PC reference space.

Nowinski and Thirunavuukarasuu analyzed 3D inconsistencies of the ventrointermediate nucleus (VIM) of the thalamus in the Schaltenbrand–Wahren atlas [25]. The 3D models were reconstructed from the axial, coronal, and sagittal microseries, respectively, by applying a shape-based method. All 3D models, placed in the SW coordinate system, were compared quantitatively in terms of location (centroids), size (volumes), shape (normalised eigen values), orientation (eigen vectors), and mutual spatial relationships (overlaps and inclusions). A significant 3D inaccuracy within each orientation was found, confirming the findings of Niemann and Nieuwenhofen [24].

As the Talairach–Tournoux print atlas was obtained from a single brain specimen, consistency in the three projections could be expected. Nevertheless, the process of atlas construction by cutting the specimen sagittally and interpolating the other two orientations manually resulted in spatial inconsistency across the orthogonal orientations. Nowinski and Thirunavuukarasuu examined consistency problem by analyzing the complete atlas simultaneously on all three orthogonal planes [26]. Two measures were introduced: consistency and discrepancy. The consistency referred to the uniformity of labeling across the orthogonal orientations and is calculated at the grid points, being the points of intersections of all three atlas planes: axial, coronal, and sagittal. The discrepancy determines the spatial offset in labeling across orientations caused by manual interpolation of the original print atlas. According to the authors’ measurements, the Talairach–Tournoux atlas has 27.4% consistency and 37.7% inconsistency, the thalamus being the most consistent structure (85.7% consistency, 5.4% inconsistency).

Accordingly, uncritical plotting of recordings into the bidimensional atlases coordinate spaces would jeopardize the consistency of the morphological and electrophysiological databases. Additionally, there is the danger that electrophysiological findings may be traced to different subnuclei of the thalamus dependent on the chosen atlas series and the different rigid or elastic matching procedures applied [27]. Understanding of this problem paved the way for more coherent atlas interpolations and constructions of consistent 3D atlases.

### 7.2. The Talairach–Nowinski System

The Talairach proportional grid system is a convenient normalization technique in clinical practice due to its straightforwardness and low computational cost. However, the original atlas has limitations. Some Talairach landmarks are not depicted on the plates, and the locations of others conflict with their definitions. The AC and PC landmarks are situated outside of their corresponding structures. Despite the AC point serving as the origin of the Talairach coordinate system, the coronal atlas plate that passes through the origin (marked as CA = 0) does not include the AC landmark. These inconsistencies in the atlas can result in errors during landmark identification and grid placement. Additionally, the L landmark on the axial plates is positioned several millimeters away from the grid, while the P landmark lies beyond the Talairach grid. Furthermore, the R landmark is entirely absent. The Talairach–Tournoux atlas does not encompass the entire Talairach space, leading to the absence of certain landmarks on various plates. For instance, the axial plates lack the S and I landmarks, the coronal plates do not have the A and P landmarks, and the sagittal plates lack the L and R landmarks. Moreover, the intercommissural plate is absent from the atlas.

The inconsistencies between the Talairach landmarks and grid can have a negative impact on the accuracy of atlas-to-data registration. To address these issues, Nowinski [28] proposed a new set of landmarks called the Talairach–Nowinski (T-N) landmarks, which provide more precise definitions and allow for more accurate identification of cortical landmarks. The T-N landmarks are defined on three planes: the intercommissural plane, and two coronal planes passing through the AC and PC, respectively. To define the intercommissural plane, Nowinski introduced several intercommissural lines that can be identified on MRI scans, including the central intercommissural line passing through the centers of the anterior and posterior commissures on the midsagittal plane, the tangential intercommissural line that runs dorso-posteriorly to the anterior commissure and ventro-anteriorly to the posterior commissure on the midsagittal plane, and the internal intercommissural line, which is the distance between the internal intercommissural landmarks (Figure 6). These definitions provide a more consistent and accurate method for identifying landmarks and allow for improved atlas-to-data registration.

The internal intercommissural distance provides the closest approximation to the original Talairach intercommissural distance with a relative intercommissural error of only 0.5%, whereas the central and tangential intercommissural distances have much higher relative intercommissural errors of about 10%. However, using the internal or central intercommissural distances can result in a high maximum displacement error at the cortex of around 11 mm, whereas the tangential intercommissural distance results in only a 1 mm error. The internal intercommissural distance is highly sensitive to the actual location of the intercommissural plane, with a sensitivity of over 10%, whereas the central intercommissural distance has very low sensitivity, below 0.5%.

As a result, when using the T-N system, it is possible to easily identify cortical landmarks using the intercommissural plane, but the selection of the intercommissural plane must be tailored to the specific application, and user control over its placement is necessary. For stereotactic and functional neurosurgery, the internal intercommissural distance is the most suitable for providing high accuracy for subcortical structures. For localisation analysis in human brain mapping research, high accuracy must be achieved at the cortex, and the tangential intercommissural line is superior.

## 8. Digital Histological Atlases

Two printed atlases currently available are the Mai atlas and the Morel atlas [29,30]. The Mai atlas, whose use is very intuitive, includes printed images and digital media [29]; a macroscopic atlas containing 17 horizontal, 15 coronal, and 8 sagittal sections is also provided. Each series of macroscopic plates is introduced with three pages of orienting diagrams depicting the horizontal, coronal, and sagittal planes, respectively. These macroscopic plates consist of macroanatomic sections accompanied by corresponding MR images and, in some cases, a bone-windowed CT image. The vascular supply of the corresponding brain territories is also provided. On the facing page, there is a comprehensively annotated artist’s tracing at the same scale to aid the reader in orientation when viewing a single page or atlas plate. Additionally, the Mai atlas includes a microscopic, myeloarchitectonic atlas, illustrating coronal sections from a single brain (shown in Figure 7). Four introductory pages diagram the orientation and location (in Talairach space) of the 69 coronal brain sections spanning a distance between 60 mm anterior to the AC to 100 mm posterior to the AC.

In 1997, Morel, Magnin, and Jeanmonod presented a microscopic stereotactic atlas of the human thalamus and basal ganglia [30]. The organization of the book was related to anatomical regions (thalamus, basal ganglia, subthalamic fiber tracts). Similarly to other atlases, data are sampled in three orthogonal planes in the AC-PC reference space. Parcellations of thalamic (Figure 8) and other basal nuclei are based on cyto- and myeloarchitectonic criteria and are further corroborated by staining for calcium-binding proteins, which bear functional significance. This information could be greatly useful to understand the mechanism of action of current surgical stimulation of basal nuclei in this region.

## 9. Three-Dimensional Atlases

The natural evolution of atlasing was the development of computerized atlases in order to overcome limitations of their print counterparts, including rigid alignment, image plate sparseness, lack or limited functionality, complicated use, lack of interactivity, and difficulty in the mapping of the atlas content into an individual brain scan [31]. This has driven the identification of 5 different solutions: (1) direct digitization of the existing print atlases [32]; (2) creation of bi-media atlases with both print and digital content [29,30]; (3) 3D extension of the existing print atlases [33,34]; (4) creation of improved atlases derived from the print content by postprocessing, enhancements, and extensions [35,36]; and (5) development of new electronic atlases (such as early ones, e.g., by Bohm et al. and Greitz et al.) constructed from digitized cryosection photographs [37,38].

### 9.1. Early Computer-Based Atlases

Probably, the first computer program with digitized (and scalable) stereotactic atlases was developed by Bertrand et al. [39]. The authors transformed map line drawings into digital data, stored in the computer memory, by tracing enlarged photographs of the transparent line drawings of the Atlas with the mechanical pen of an analogue X Y plotter. The drawings obtained could then be displayed on the viewing screen of a Tektronix type 4002 ‘Computer Graphics Terminal’. The software could be interrogated using a number of questions, the answers to which were readily available from simple measurements on the stereotaxic ventriculogram: the distance between the commissures, the height of the thalamus above the intercommissural plane, and, if visible, the width of the third ventricle. Answers to these questions were simply typed on the terminal keyboard. They are used to compress or expand the corresponding dimensions of the map to match the individual patient’s brain, at least as far as the X-ray landmarks are concerned.

A digital version of the Schaltenbrand and Wahren atlas resident in a computer was created by Kall et al. [32]. The authors, for the first time, were able to digitalize the atlas and to warp it onto CT scans of individual patients using polar coordinates. The CT database was reformatted in planes having the same orientation as horizontal microscopic sections of a computer-based stereotactic atlas. The software was able to align the atlas to fit within computer-based anatomical boundaries by polar scaling. Thus, the reformatted CT slice became a labeled atlas of the individual patient’s brain.

### 9.2. Atlas-to-Scan Warping

While the Talairach atlas remains the most commonly used system for reporting coordinates in neuroimaging studies, the absence of an actual 3D image of the original brain used in its construction has severely limited the ability of researchers to automatically map locations in 3D anatomical images [40].

Warping atlases to pre-operative patient data has been one solution to solve the issue of lack of three-dimensionality of traditional printed atlases and to obtain individual functional data directly on a patient 3D MRI. Registration of the atlas to the patient is typically achieved using linear scaling techniques based primarily on the length of the AC–PC line and the width of the third ventricle. Reliable atlas-to-brain warping, however, requires setting multiple parameters and complex software packages developed to this purpose. They differ not only in functionality provided but also in brain normalization methods applied and brain atlas, the Talairach and Tournoux atlas and/or the MNI template being typically employed. The two most common methods currently used for warping were: (1) a direct, piecewise linear scaling of each individual subject into Talairach space such as that which is applied in, for example, the AFNI software package or (2) mapping each individual subject into a common reference space (the most common reference space is the Montreal Neurological Institute (MNI) space) and then applying a piecewise linear conversion for mapping MNI coordinates to Talairach coordinates [41,42].

Nowinski et al. developed a digital atlas that incorporated data from three print atlases, including Ono et al., 1990; Schaltenbrand and Wahren, 1977; and Talairach and Tournoux, 1988 [35,43]. In order to register the atlas to a subject or patient, a piecewise linear approach was used to transform the atlas to the MR volume. The Talairach transformation (TT) is, at present, the most widespread method for brain normalization and atlas-to-data warping [44]. Conceptually, it is simple from an anatomy and transformation standpoints. The TT is based on the eight Talairach point landmarks: anterior commissure (AC) and posterior commissure (PC) located on the midsagittal plane (MSP), and six cortical landmarks determining the extents of the brain in the anterior (A), posterior (P), left (L), right (R), superior (S), and inferior (I) directions. The transformation warps the source volume image into the target image piecewise linearly with 13 degrees of freedom (DOFs).

The described methods have the limit of accounting only for the overall size and orientation of the brain, but not for any other variable. It has been shown that the overall shape of the MNI template is taller, longer, and has larger temporal lobes than the Talairach brain. Brett et al. created the mni2tal transform that mapped MNI space into Talairach space using two linear transformation matrices but noted that the technique was merely an approximation [45]. To improve on the above-mentioned techniques, researchers have studied the disparity between Talairach coordinates derived from different methods and have applied various affine transforms to optimize the coordinates between MNI space and Talairach space [46,47,48]. While affine transformations contain more information than the piecewise-linear transformations, they still do not account for differences in brain shape. Non-linear registrations employ thousands of parameters instead of the 12 parameters used typically for affine registrations and, therefore, can account for these regional shape differences. Theoretically, nonlinear and high degree-of-freedom methods for warping the brain with a high degree of freedom (DOF) are considered more effective and precise than those using low/medium DOFs and piecewise linear methods. However, accuracy alone is not the only determining factor for the widespread adoption of brain warping methods. Other important considerations include execution time, user acceptance, ease of use, validation, and accessibility.

Nowinski et al. [44] developed the Fast Talairach Transformation (FTT) method, which enabled the automatic and near-real-time registration of a brain atlas to neuroimages. By automatically calculating landmarks and warping the Talairach and Tournoux atlas, the FTT method only took approximately 5 s on a standard computer. After registering the individualized atlas to the patient-specific brain scan, the scan could be labeled, segmented, and searched for a given structure using the atlas. The FTT method was validated for 215 MR scans and demonstrated a localization accuracy of about 1 voxel for the AC and PC and half a voxel for the cortical landmarks.

Digitizing atlases offers several benefits, including fast planning suitable for surgical procedures, increased targeting accuracy through the use of multiple orientations and atlases, and contour representation superimposed on the data, as well as global and local registrations using any clearly visible landmarks. Real-time interactive atlas warping is feasible at any time, and the neurosurgeon can plan more sophisticated trajectories by displaying the trajectory on all three planes and in 3D. Additionally, digitizing atlases can reduce the invasiveness and risk of the surgical procedure by reducing the number of microelectrodes necessary for exploration [23].

### 9.3. Digital Atlases from MRI-Data

In the late 1980s, the introduction of MRI offered the opportunity for the introduction of in vivo brain structure localization. The use of MRI also stimulated the production of conceptually new, fully three-dimensional brain atlases. These types of atlases are useful because they allow easy three-dimensional navigation in the brain (Figure 9).

The Surgical Planning Laboratory (SPL) at MGH, Harvard Medical School in the USA embarked on a project in 1990 to create a detailed anatomical brain atlas using a T1-weighted MRI. The primary objective of the project was to create an educational tool, now known as the SPL anatomy browser, which can could be accessed online at no cost (www.spl.harvard.edu, accessed on 1 May 2015). This browser enables users to navigate through a T1-weighted MRI volume containing various cortical and subcortical structures. A similar approach was adopted also by our group [49]. The atlas based on MRI was structured into four categories of picture sections: three types of sections were obtained from the same brain and were oriented in the standard spatial planes of axial, sagittal, and coronal, while the fourth type comprised three-dimensional pictures generated by computerized processing of the previous pictures (Figure 10). While the authors did not assert that the atlas had the same level of precision as other atlases of its kind, the magnetic resonance-based organization and life-size tables made it akin to a traditional stereotactic atlas.

Many groups have developed their homemade MRI-based 3D brain atlases and included them in brain segmentation methods. Despite the advantages of direct anatomic localization, MRI is limited in terms of spatial resolution, which makes these approaches less accurate in the context of DBS clinical practice.

## 10. Three-Dimensional Rendered Atlases from Histological Sections and Multiatlas Collections

Histology remains the most effective tool for revealing the anatomy of the basal ganglia, although it can only provide two-dimensional images of the brain. Efforts have been made to create 3D atlases based on histological sections. Creating a 3D atlas from 2D histological sections requires a method for aligning the entire series of individual sections into a consistent 3D block. Technically, it is feasible to interpolate image or contour data from sparse atlases, which allows for the reconstruction of 3D models and the generation of intermediate sections. For example, the Schaltenbrand–Bailey atlas was interpolated with a 0.5 mm step [50]. The Cerefy series is the most commonly used collection of computerized atlases. The Cerefy anatomical atlases have been derived from the classic print brain atlases edited by Thieme [21,22,43,51]. To create these computerized atlases, the original print materials were processed as follows: (1) scanning of images, (2) full segmentation (contouring or color coding) of all atlas structures, (3) complete labelling (naming) of all atlas structures, (4) arrangement of the atlas images into volumes, (5) atlas checking, correcting, enhancing, and extending, (6) constructing 3D versions, (7) developing various representations in 2D and 3D (bitmap, contour, polygonal, and volumetric), and (8) mutual coregistration of all 2D and 3D atlases [2,52,53,54,55,56]. Despite the high quality of those procedures, interpolation or 3D modeling do not compensate for the intrinsic shortcomings of the original print material. An analysis of the main target structures in the Schaltenbrand–Wahren atlas reconstructed in 3D showed that their shapes were not fully realistic [57].

In 2007, a group from the Hopital Salpêtrìere in Paris utilized a novel method to construct 3D surfaces of the basal ganglia and thalamus. This involved using a large number of sections (160) with a 0.35 mm interval along with MRI acquisitions of the same specimen. This approach differed from that used for the Talairach atlas, which involved tracing contours from a single sagittal section (with a 4 mm interval) and extrapolating them to the coronal and axial planes via point-to-point projection [58]. Similarly, the Schaltenbrand–Wahren atlas included three series of sections (sagittal, coronal, and axial), but the number of sections was low (18, 20, 20), the section interval was high and variable (ranging from 1 to 4 mm), and the 3D coherency appeared limited. The atlas developed by Collins and colleagues was also based on serial histological data, with a section interval of 0.7 mm and 86 pairs of slices. Contours of the basal ganglia and thalamus were traced from histological sections and digitized, and a multimodal optimization (MRI, cryoblock, nissl, calbindin) and a 3D optimization procedure were implemented to ensure optimal 3D coherency.

Neurosurgical workstations commonly feature computerized brain atlases, such as the Cerefy Electronic Brain Atlas Library and/or Cerefy Brain Atlas Geometrical Models. These atlases can be found in various systems, including the StealthStation (Medtronic Surgical Navigation Technologies), Target and iPlan (BrainLAB AG), SurgiPlan (Elekta Instrument), SNN 3 Image-Guided Surgery System (Surgical Navigation Specialists), and the neurosurgical robot NeuroMate (Integrated Surgical Systems). Additionally, the Cerefy brain atlas libraries are being evaluated by companies such as Prosurgics, Renishaw, Cedara Software, and Z-KAT. Other companies, such as Tyco/Radionics and Stryker/Leibinger, have also developed their own digital versions of the SWand TT print atlases. The COMPASS System of Stereotactic Medical Systems and the CASS system of MIDCO also offer electronic atlases.

## 11. Probabilistic and Functional Atlases

An alternative method of atlasing involves creating an atlas based on functional data obtained from a group of subjects or patients [55]. This data can consist of preoperative electrophysiological recordings or clinical data, such as points that elicit movement disorders or sensory perceptions, or postoperative tuning data, such as a contact that triggers a specific sensation.

To improve the accuracy of surgical targeting, various researchers have collected and analyzed functional data from subcortical structures of multiple individuals during stereotaxy [59,60,61,62]. The data were transformed into alphanumeric codes to standardize the electrophysiological data, which were then normalized to an anatomical atlas to create composite functional maps. By registering electroanatomic observations from multiple patients to a common coordinate space, it became possible to examine functional organization in relation to anatomical structures. Bertrand et al. first introduced this technique by displaying a rough somatotopic organization of the corticobulbar and corticospinal fibers in the posterior limb of the internal capsule, derived from a group of 26 patients normalized to a representative plate of the Schaltenbrand Bailey atlas [63]. Later, they expanded this technique to include interactive recording and display of physiological responses collected during surgery [59]. In 1982, Tasker et al. conducted microstimulation on 9383 sites during 198 procedures, primarily for Parkinson’s disease and chronic pain, providing one of the most comprehensive analyses of electrophysiological observations obtained through microstimulation to date [60]. Finnis et al. non-linearly registered electrophysiological data obtained from 88 patients (106 procedures) via microelectrode recording and electrical stimulation to the patient’s MRI and a high-resolution MRI reference brain [64]. The authors found clustering of interpatient physiologic responses within the thalamus, globus pallidus, subthalamic nucleus, and adjacent structures. These data were then registered to a three-dimensional patient MRI within an image-guided visualization program that enabled prior delineation of surgical targets, anatomy with a high probability of containing specific cell types, and functional borders. This method’s advantages included non-linear registration to accommodate interpatient anatomical variability and avoidance of digitized versions of printed atlases of anatomy as a common database coordinate system. Nowinski and Benabid have collaborated on an atlas, referred to as the ‘probabilistic and functional atlas’ (PFA), which is based on clinical data gathered from 274 patients with Parkinson’s disease who underwent surgery at the Joseph Fourier University School of Medicine in Grenoble, France [65,66,67]. This atlas combines intraoperative neuroelectrophysiology, pre- and intra-operative neuroimaging studies, and postoperative neurological assessments to produce probabilistic functional maps, which are generated by converting the coordinates of the neurologically most effective contacts into functional maps. Additionally, the authors incorporated the patient-specific anatomy and geometry of a stimulating electrode into their approach. This method yields a quantitative spatial distribution of the best stereotactic targets in a normalized atlas space, while also allowing for the study of functional properties of structures and determination of targeting accuracy. The PFA is a volumetric atlas, consistent in 3D, and it can be reformatted in any orientation. The PFAs were created for the subthalamic nucleus (STN) and ventral intermediate nucleus (VIM) with a high spatial resolution of 0.25 mm and accuracy of 0.25 mm (Figure 11).

An Internet portal for stereotactic and functional neurosurgery combined with a PFA was also developed in which any neurosurgeon was able to query about best targets as well as input his or her own microrecordings, convert them into probabilistic functional maps, and merge them if needed with the maps of other neurosurgeon [68]. Thus, the PFA overcomes the limitations of existing computerized brain atlases, whereas the public availability enabled to construct a progressively more accurate PFA of the human brain for standard and future stereotactic targets by the neurosurgical and neuroscience communities. Although this portal is no longer in use, sharing of the PFA through community-based portals and the progressive implementation by neurosurgeons represent a paradigm shift from a manufacturer-centric to a community-centric approach to stereotactic and functional neurosurgery.

A further step in the creation of functional atlases was proposed by Haegelen et al. collecting functional data not only of the electrode location, but also the electrical distribution of the current, represented by the volume theoretically activated by each stimulation [69]. In order to model the electric field, the authors used a pre-defined 3D Gaussian. Based on the lead’s characteristics, a monopolar stimulation and on the quasi-static potential equation, the stimulation influence covered approximately a 3 mm radius sphere around each stimulation contact. As a common space, authors chose a multi-subject MR template created from a population of patients with Parkinson’s Disease, named the ParkMedAtlis template.

## 12. Brain Atlas Applications

Atlas-based applications and future brain-related developments have been thoroughly reviewed by Nowinski [31,70]. Indeed, we can envisage three major fields of applications.

### 12.1. Education

Although technological advancements have opened up new possibilities for brain atlasing, they can also result in higher costs and reduced accessibility for users in less privileged countries. To tackle this issue, Nowinski created a 3D neuroimage public repository called NOWinBRAIN, which can be found at www.nowinbrain.org (accessed on 18 May 2023). It is the largest and most comprehensive repository to date, consisting of over 7800 images (version 3.1) organized into 10 galleries (Figure 12), featuring various novel features such as multi-tissue galleries, spatially co-registered image sequences, and unique image-naming syntax. It is an easily accessible web resource without any password or registration, making it particularly valuable for neuroeducators, medical students, neuroscientists, and clinicians, especially those in less privileged countries [71,72].

### 12.2. Research

Brain atlases have become an essential tool in modern neuroimage analysis, particularly in human brain mapping research. Brain atlases, such as BrainMap and Brain Atlas for Functional Imaging, provide the necessary neuroanatomical information to automatically label cortical areas and stereotactic coordinates in functional images [73,74,75,76,77,78]. They are also widely used for fast and robust segmentation of neuroimages. Additionally, brain atlases enable the integration of various brain-related information, such as micro- and macrostructural parcellation, connectivity, and functional specialization, which is critical for data integration. Brain atlases are also useful for localizing experimental data and planning experiments, as well as generating hypotheses about brain organization. Moreover, brain atlases facilitate knowledge discovery, as evidenced by Makowski et al.’s study, which employed genetically informed brain atlases to uncover the impact of genetic variants on brain development and neuropsychiatric risk [79]. This study identified 440 genome-wide significant loci associated with early neurodevelopment and neuropsychiatric risk in about 40,000 adults and 9000 children.

### 12.3. Clinical Applications

Human brain atlases have been widely used in clinical applications, particularly in stereotactic and functional neurosurgery. Initially, digital atlases such as The Electronic Clinical Brain Atlas were used offline in the operating room and later incorporated into surgical workstations such as the StealthStation to assist neurosurgeons in pre-, intra-, and post-operative support [52]. Brain atlases are useful for planning target and trajectory, identifying structures intersected by the trajectory, and specifying the structures already traversed by the electrode. They also measure distances to important structures, provide neuroanatomic and vascular context, and examine the precision of electrode placement. While current targeting techniques focus on the target point, atlas reporting cerebrovascular anatomy may assist in better placement of the entry point. Indeed, a small change in the entry point of a DBS electrode, occurring quite commonly especially in frameless DBS procedures, can lead to a major change in the trajectory. Nowinski et al. created an atlas of structure and vasculature with >900 vessels, with the smallest being 90 μm in diameter (Figure 13), it can be used to analyze the track–brain spatial relationship, allowing the DBS electrode to be placed more effectively, potentially reducing the invasiveness of the DBS procedure [80].

Brain atlases are also employed in neuroradiology, where they can assist in neuroimage interpretation by segmenting and labeling brain scans including pathological reporting, facilitating processing of multi-detector scans, and enabling communication between physicians and patients. Brain atlases also have potential in stroke management, neurology, and psychiatry [81,82,83]. For example, they can provide automated processes for predicting, diagnosing, and treating strokes, demonstrate various locations of brain damage and resulting neurologic deficits, and generate neuroanatomic volumes of interest for statistical analysis of psychiatric disorders such as schizophrenia [84].

## 13. Ongoing Projects

In our era, in which sophisticated anatomical and functional imaging can be easily obtained for each of our patients, anatomical atlases based on histological and histochemistry data from cadaver specimens have lost their important function of being an exclusive instrument for deep brain areas localization. However, they still represent a distinctive instrument for understanding and learning the anatomy of basal ganglia and their mutual spatial relationships. Furthermore, the combination of anatomic information together with electrophysiology, cytoarchitectural, connectomics, and molecular findings provides a unique knowledge of the human brain as a whole [85]. In recent years, there has been a significant increase in human-brain-related projects, initiatives, and programs. These efforts, which are often large, advanced, government-led, and/or well-funded, include: The Human Connectome Project, the Allen Brain Atlas, the Big Brain, the CONNECT project, the Brainnetome project, the BRAIN Initiative, the Blue Brain Project, the Human Brain Project, the Chinese Color Nest Project, the Japanese Brain/MINDS project, and SYNAPSE (Synchrotron for Neuroscience—an Asia-Pacific Strategic Enterprise) [86,87,88,89,90,91,92,93,94,95,96]. These projects aim to map the structural and functional connections of the brain, understand brain circuits and behavior, develop technology to advance neuroscience discovery, simulate neocortical micro-circuitry, and create brain-inspired information technology. As a result of these efforts, there is now a wealth of big data and diverse brain-related databases, including BigBrain, Allen Brain Atlas, HCP database, BIRN MRI and fMRI data, OpenNEURO, OASIS Brains Project, ABCD Data Repository, BCP database, BP neuroimaging database, and the Alzheimer’s Disease Neuroimaging Initiative [97]. These databases have enabled the acquisition of vast amounts of data and have provided the basis for the development of novel and more powerful brain atlases. The use of high-performance computing at the peta- and exascale will further enhance our understanding of the human brain at various scales [70].

## 14. Conclusions

Brain atlasing has evolved significantly from the early days of hand-drawn cortical maps to sophisticated brain atlas platforms. Today, human electronic brain atlases have made great strides in terms of their content, functionality, and applications. Progress in this field has been driven by the use of software engineering methods and tools, such as databases, image processing, computer graphics, and virtual and augmented reality. This progress has been made in various areas, including scope, parcellation, plurality, modality, scale, ab/normality, ethnicity, and their combination. It is predictable that the large-scale brain projects and the accumulation of significant amounts of data will have a tremendous impact on neuroanatomy and brain atlasing.

## Figures and Tables

**Figure 1 brainsci-13-00830-f001:**
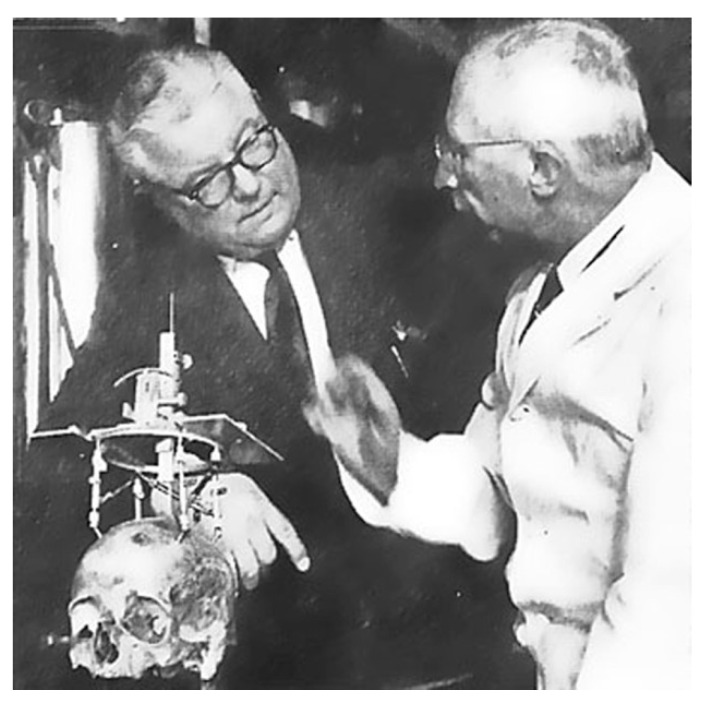
Spiegel and Wycis with their stereotactic apparatus.

**Figure 2 brainsci-13-00830-f002:**
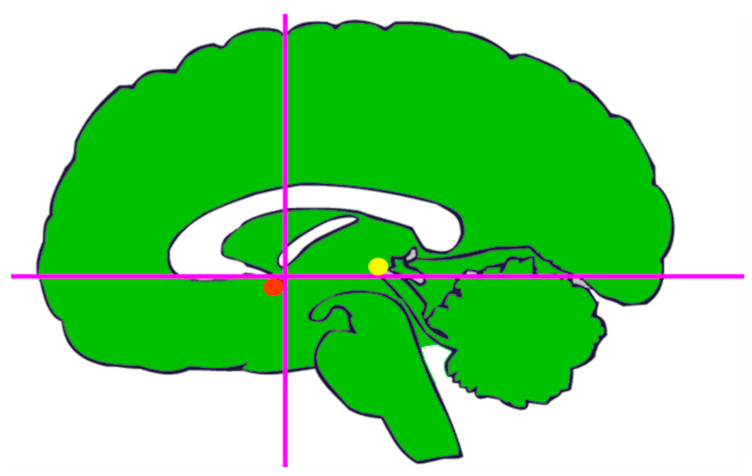
The intercommissural line (CA-CP-line) passes through the superior edge of the anterior commissure (red dot) and the inferior edge of the posterior commissure (yellow dot). The vertical line (VCA) passes through the posterior margin of the anterior commissure. These lines could be drawn directly on the ventriculograms.

**Figure 3 brainsci-13-00830-f003:**
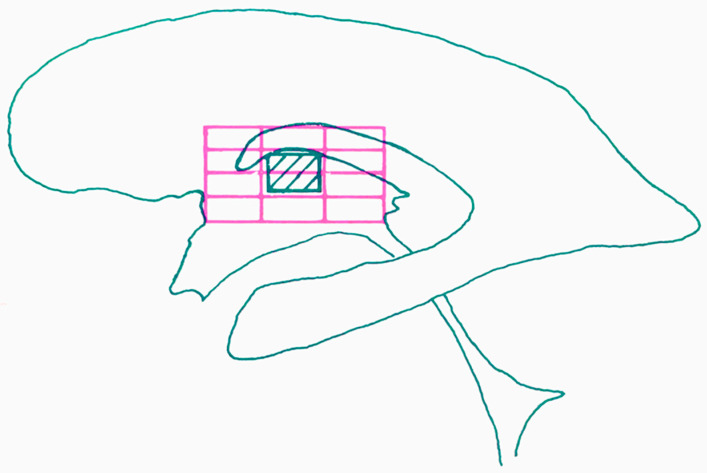
Reconstruction of the ventro-lateral nucleus of the thalamus in the lateral projection using the proportional system by Talairach. The dashed rectangle corresponds to the ventro-lateral nucleus of the thalamus. Schematic representation of the superimposition of the Talairach’s diagram (purple grid) over the patient’s ventriculogram.

**Figure 4 brainsci-13-00830-f004:**
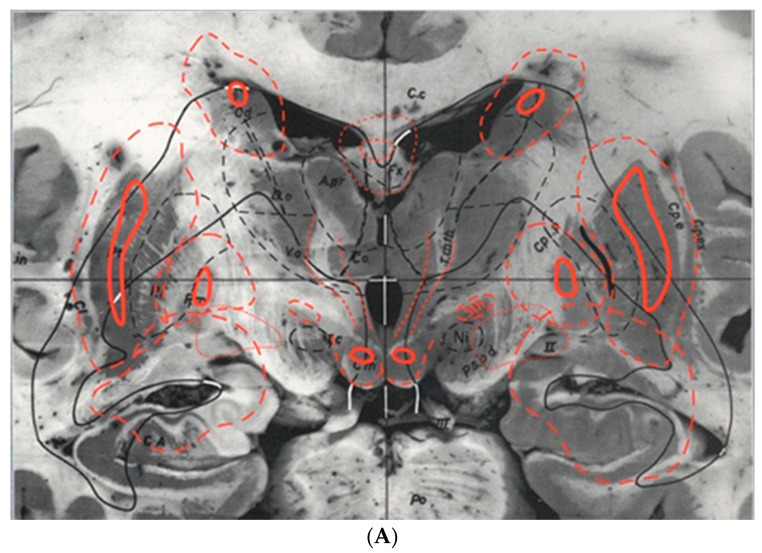
(**A**). Plate five from the Schaltenbrand–Wahren brain atlas showing the basal nuclei (from the Schaltenbrand and Wahren atlas, reproduced with permission; copyright Thieme: Stuttgart, Germany, 1977)) [21]. (**B**). Plate 43, brain LXXVIII, myelin-stained sagittal Section 12.0 mm from the midline. It is likely that this particular atlas section has been used to guide most stereotactic operations for movement disorders involving the subthalamic nucleus region in the modern era of MR-image-guided deep brain stimulation for Parkinson’s disease.. (from the Schaltenbrand–Wahren brain atlas, reproduced with permission; copyright Thieme: Stuttgart, Germany, 1977) [21].

**Figure 5 brainsci-13-00830-f005:**
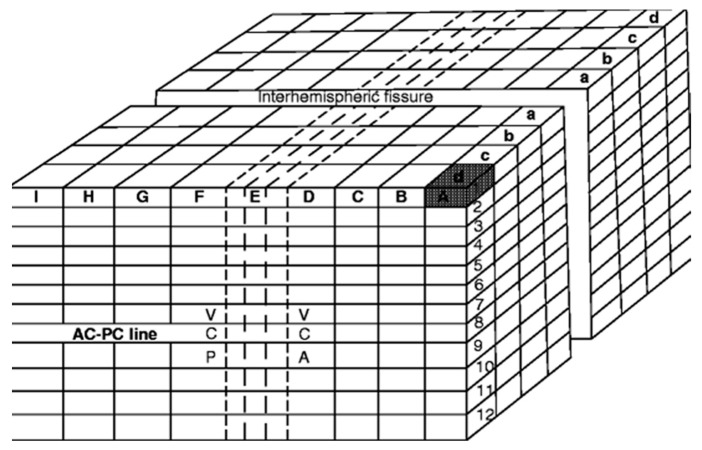
The Talairach’s space, represented by orthogonal rectangular prisms (‘‘parallelograms’’) encompassing the brain. Each sub-volume in the brain is identified by three dimensions that correspond to the principal axes of the brain, and these dimensions are represented by a capital letter, a lowercase letter, and a number, respectively. For example, the shaded area in the upper right-hand front corner can be identified as A-d-1 (information adapted from Talairach and Tournoux with permission; copyright Thieme: Stuttgart, Germany, 1988) [22].

**Figure 6 brainsci-13-00830-f006:**
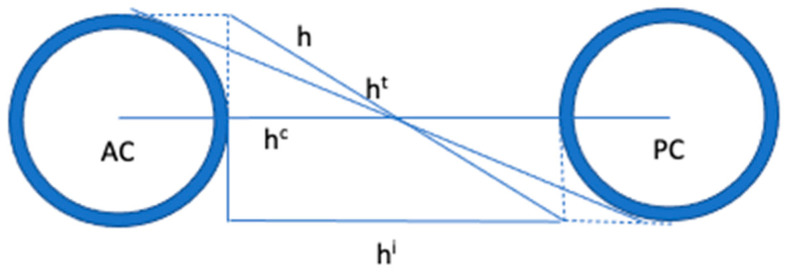
To define the intercommissural plane, different intercommissural lines are introduced to be defined on MRI. (h): original intercommissural distance; (hc): central intercommissural distance between the central intercommissural landmarks; (hi): internal intercommissural distance between the internal intercommissural landmarks; (ht): tangential intercommissural distance between the tangential intercommissural landmarks (modified from Nowinski [28]).

**Figure 7 brainsci-13-00830-f007:**
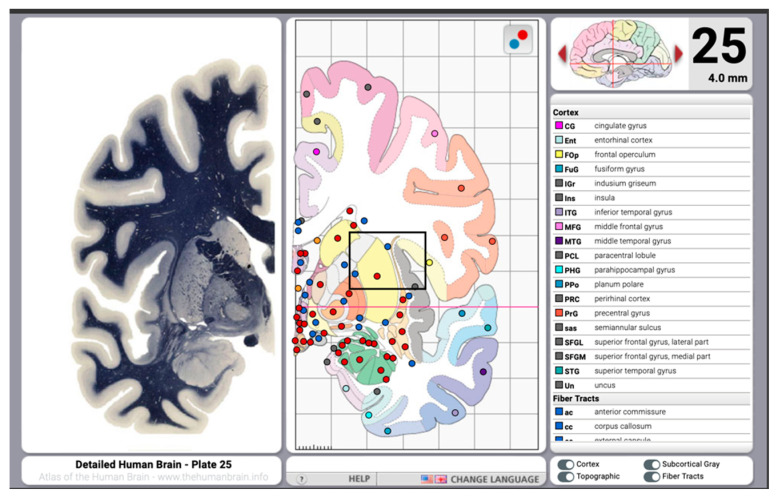
Macroanatomic sections accompanied by a comprehensively annotated artist’s tracing at the same scale (from the freely available web-based version of the atlas): http://www.thehumanbrain.info/brain/bn_brain_atlas/brain.html, accessed on 18 May 2023).

**Figure 8 brainsci-13-00830-f008:**
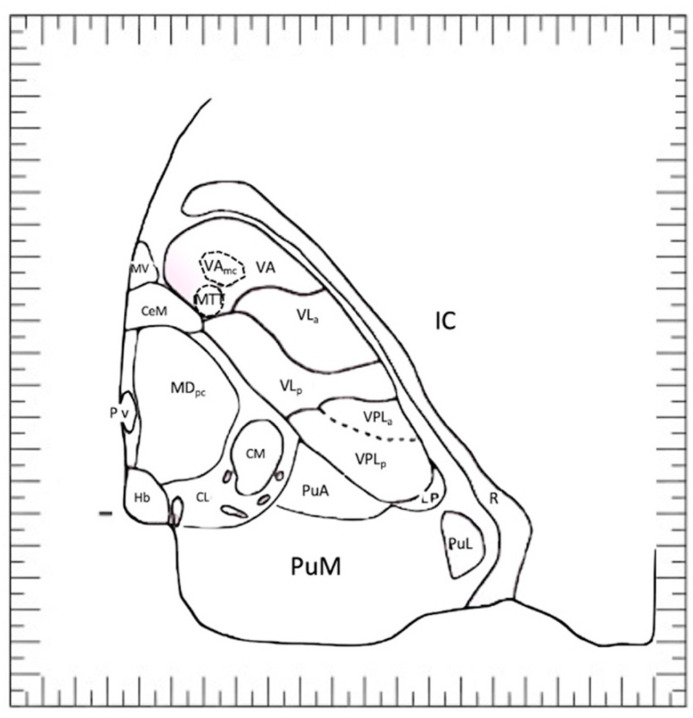
Nuclear anatomy of thalamus according to the thalamic atlas of Morel^10^. **CeM**, central medial; **CL**, central lateral nucleus; **CM**, center median nucleus; **Hb**, habenular nucleus; **LD**, lateral dorsal nucleus; **LP**, lateral posterior nucleus; **MDpc** and **MDpl**, mediodorsal nucleus, parvocellular and paralamellar divisions; **MTT**, mammillothalamic tract; **MV**, medioventral; **PuA**, anterior pulvinar; **PuL**, lateral pulvinar; **PuM**, medial pulvinar; **Pv**, paraventricular; **R**, reticular nucleus; **VA** and **VAmc**, ventral anterior nucleus and magnocellular division; **VLa**, ventral lateral anterior nucleus; **VPLp** and **VPLa**, ventral posterior lateral nucleus, posterior and anterior divisions.

**Figure 9 brainsci-13-00830-f009:**
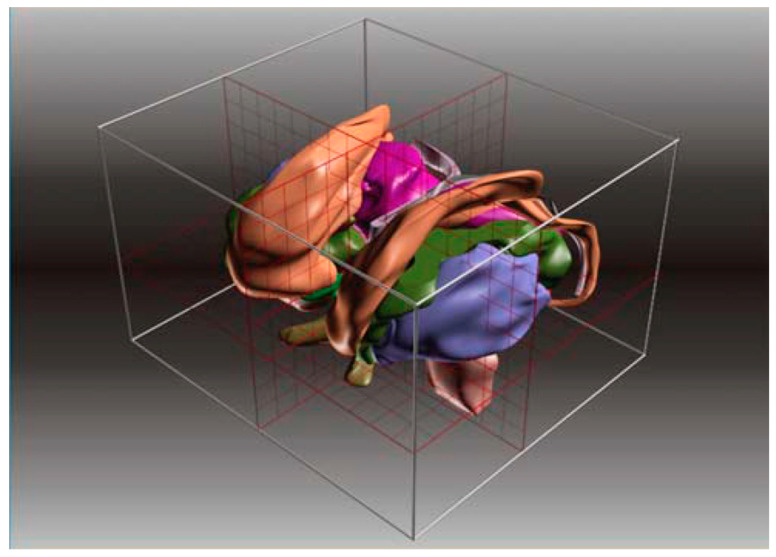
Three-dimensional spatial limits in the 3D-MRI-based Atlas by Lucerna et al. [49] (reprinted with permission; copyright: Springer: Wien, 2002).

**Figure 10 brainsci-13-00830-f010:**
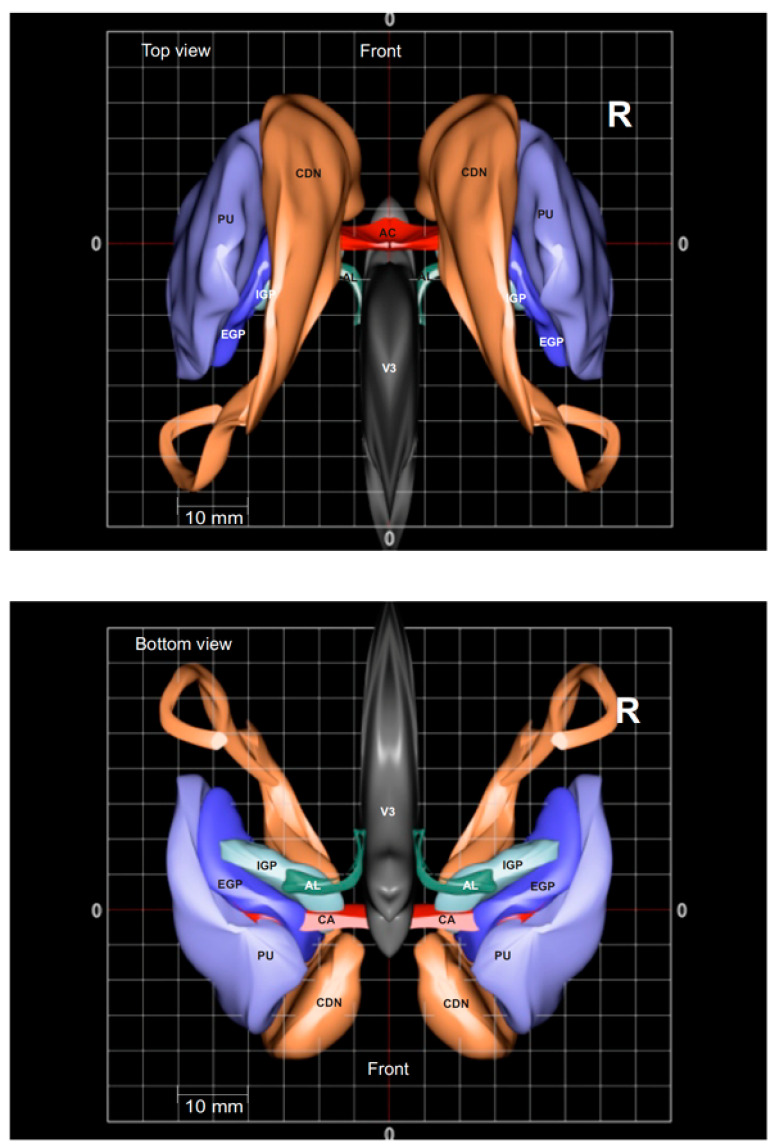
Three-dimensional rendering of the caudate nucleus (CDN), Putamen (PU), globus pallidus externus (GPE), globus pallidus internus (GPI), ansa lenticularis (AL) in the 3D-MRI-based Atlas by Lucerna et al. [49] (reprinted with permission; copyright: Springer: Wien, 2002).

**Figure 11 brainsci-13-00830-f011:**
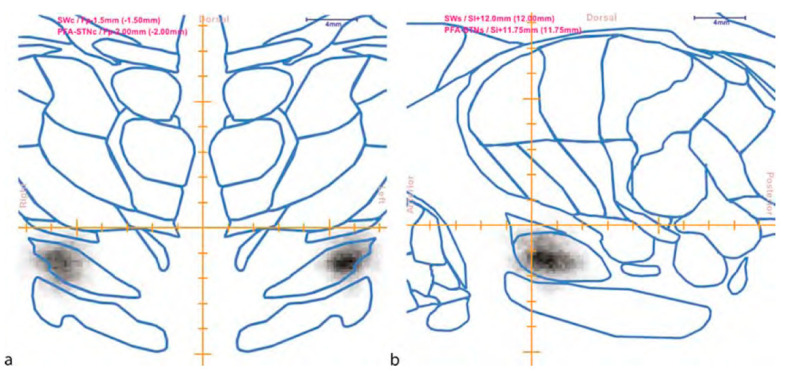
A combined Shaltenbrand and Wahren Atlas (SWA) Probabilistic Functional Atlas (PFA) identification of the subthalamic nucleus ((**a**): coronal view; (**b**): sagittal view). The PFA is presented in gray scale, with a gray level proportional to probability. The SWA is displayed as blue contours. The coordinates of the atlas images are shown in the top left corner. (Reproduced from Nowinski with permission; copyright: Springer, Berlin/Heidelberg, 2009) [55].

**Figure 12 brainsci-13-00830-f012:**
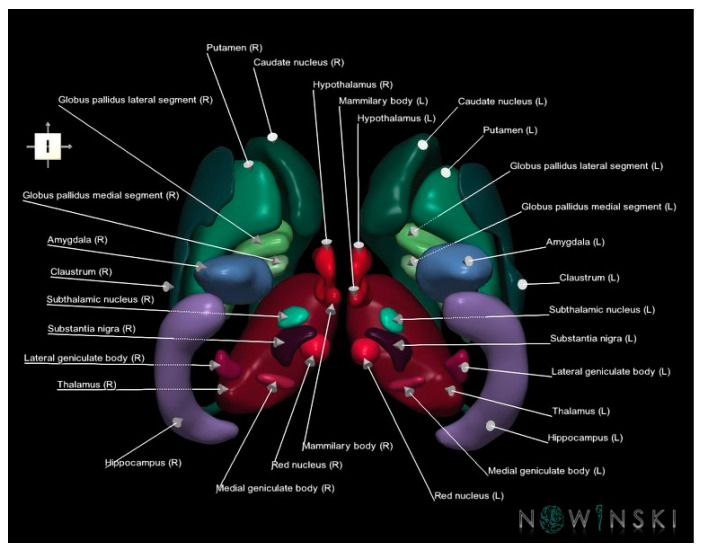
Three-dimensional rendering of the basal ganglia from the 3D neuroimage public repository called NOWinBRAIN (www.nowinbrain.org, accessed on 18 May 2023).

**Figure 13 brainsci-13-00830-f013:**
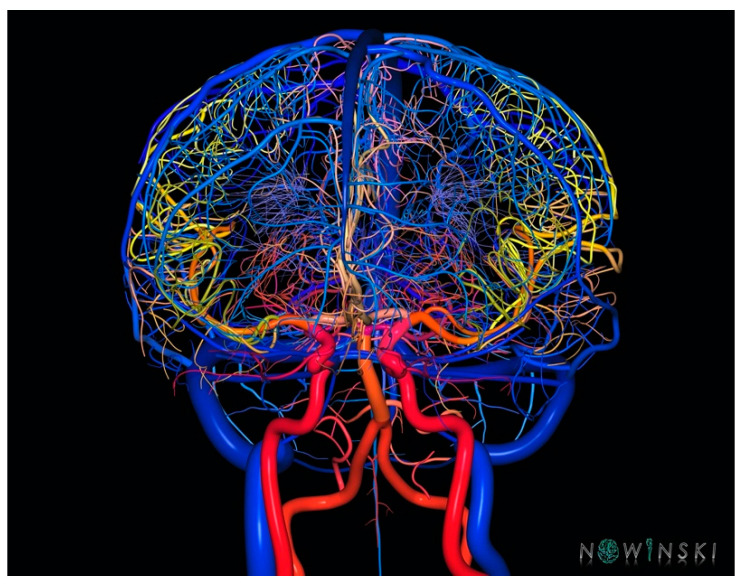
A three-dimensional rendering of arterial and venous structures. Three-dimensional atlas of human vasculature can be used to analyze in DBS to analyze track–brain spatial relationship allowing the DBS electrode to be placed more effectively (from www.nowinbrain.org, accessed on 18 May 2023).

## Data Availability

Not applicable.

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
