# Peer review of "A Brief History of Stereotactic Atlases: Their Evolution and Importance in Stereotactic Neurosurgery"

_brainsci, 2023, doi:10.3390/brainsci13050830_

Round 1

Reviewer 1 Report

Comments and Suggestions for Authors

The introduction presents a clear overview of the importance of stereotactic techniques in neurosurgery, explaining how they enable the localization of deep-sited brain regions and hidden brain areas with no direct surgical exposure. The discussion on brain atlases provides a useful historical context, highlighting the key contributions of researchers in this field. The subsequent sections provide a detailed historical account of brain localization, craniocerebral topography, and the development of stereotactic techniques. The information presented is well-organized, and the writing style is clear and easy to follow. Overall, the text appears well-researched and well-written, providing a comprehensive overview of the history of brain atlases and their importance in neurosurgery.

Here are some recommendations:

1)    Page 1, Lines 44-45: Although atlas-based information is essential, you also need artifact-free MR images for both direct targeting and atlas-based targeting. Most of the time these two acts as complementary to each other. So, you also cannot rely on atlas-based targeting in case of imaging artifacts.

2)    Page 2, Line 87: I believe “radiosurgery” should be “neurosurgery”.

Author Response

Reviewer Comments: The introduction presents a clear overview of the importance of stereotactic techniques in neurosurgery, explaining how they enable the localization of deep-sited brain regions and hidden brain areas with no direct surgical exposure. The discussion on brain atlases provides a useful historical context, highlighting the key contributions of researchers in this field. The subsequent sections provide a detailed historical account of brain localization, craniocerebral topography, and the development of stereotactic techniques. The information presented is well-organized, and the writing style is clear and easy to follow. Overall, the text appears well-researched and well-written, providing a comprehensive overview of the history of brain atlases and their importance in neurosurgery.

Response: We express our gratitude for the reviewer's favorable assessment of our manuscript. Based on multiple comments from the reviewers, the revised manuscript is substantially different from the previous version, with a great deal of new points and different aspects analyzed. We sincerely hope that our efforts can be recognized and that the new manuscript considered a further improvement.

Here are some recommendations:

Point 1)    Page 1, Lines 44-45: Although atlas-based information is essential, you also need artifact-free MR images for both direct targeting and atlas-based targeting. Most of the time these two acts as complementary to each other. So, you also cannot rely on atlas-based targeting in case of imaging artifacts.

Thank you. This point has been addressed

2)    Page 2, Line 87: I believe “radiosurgery” should be “neurosurgery”.

Thank you. This point has been addressed

Reviewer 2 Report

Comments and Suggestions for Authors

This paper nicely presents a brief history of stereotactic atlases. Its major limitation is that it focuses predominantly on early developments and is not complete. This work can benefit from the following extensions and enhancements.

1. Provide a section on early electronic atlases, in particular the pioneering work done by Bertran et al Keely et al.

Bertrand G, Olivier A, Thompson CJ. (1974). Computer display of stereotaxic brain maps and probe tracts. Acta Neurochir Suppl 21:235-243.

Kall BA, Kelly PJ, Goerss S, Frieder G. (1985). Methodology and clinical experience with Computed Tomography and a computer-resident stereotactic atlas. Neurosurgery;17(3):400-407.

2. Discuss in more detail the AC and PC

In particular:

·       The Talairach AC point landmark is beyond the AC structure, and it is missing in the Talairach 1988 atlas at the coronal CA=0 plate i.e. at the center of the coordinate system (in my version of the atlas Prof. Talairach added this landmark manually).

·       Although the Talairach and Schaltenbrand atlas are based on the AC and PC, these landmarks are defined differently.

·       On MRI patients’ scans used for surgery planning the AC and PC are usually defined as center points.

·       Various definitions and AC and PC landmarks and the resulting differences are addressed e.g., in

Nowinski WL: Modified Talairach landmarks. Acta Neurochirurgica 2001;143(10);1045-1057.

3. It may be worth mentioning about automated methods for the calculation of the Talairach landmarks and atlas-to-scan mapping, e.g.

Nowinski WL, Qian G, Bhanu Prakash KN, Hu Q, Aziz A: Fast Talairach Transformation for magnetic resonance neuroimages. Journal of Computer Assisted Tomography 2006;30(4):629-41.

4. Elaborate more about properties, especially limitations, of the Talairach and Tournoux 1988 and the Schaltenbrand and Wahren atlases, e.g.:

Niemann K, Naujokat C, Pohl G, Wollner C, von Keyserling KD (1994) Verification of the Schaltenbrand and Wahren stereotactic atlas. Acta Neurochir (Wien) 129(1–2):72–81.

Niemann K, van Nieuwenhofen I (1999) One atlas-three anatomies: relationships of the Schaltenbrand and Wahren microscopic data. Acta Neurochir (Wien) 141:1025–1103.

Nowinski WL, Thirunavuukarasuu A: Quantification of spatial consistency in the Talairach and Tournoux stereotactic atlas. Acta Neurochirurgica 2009 151(10):1207-13.

Nowinski WL, Liu J, Thirunavuukarasuu A: Quantification and visualization of three-dimensional inconsistency of the ventrointermediate nucleus of the thalamus in the Schaltenbrand-Wahren brain atlas. Acta Neurochirurgica 2008;150(7):647-53.

5. Discuss in more detail the differences between the Schaltenbrand and Bailey atlas (claimed by the authors to be “probably the world’s most widely used one”) and the Schaltenbrand and Wahren atlas (to be the most prevalent in surgical workstations).

6. The way of the atlas use is vaguely addressed.

In particular:

·       The atlas can be used preoperatively, intraoperatively, and postoperatively, for instance

Nowinski WL: Computerized brain atlases for surgery of movement disorders. Seminars in Neurosurgery 2001;12(2):183-194.

·  This survey addresses targeting, but the atlas can be used for entry point localization, for instance

Nowinski WL, Chua BC, Volkau I, Puspitasari F, Marchenko Y, Runge VM, Knopp MV: Simulation and assessment of cerebrovascular damage in deep brain stimulation using a stereotactic atlas of vasculature and structure derived from multiple 3T and 7T scans. Journal of Neurosurgery 2010;113(6):1234-41

·  Address the development of electrophysiology databases, e.g.,

Finnis K, Starreveld Y, Parrent A, Sadikot A, Peters T. (2003). Three-dimensional database of subcortical electrophysiology for image-guided stereotactic functional neurosurgery. IEEE Trans Med Imag.;22(1):93–104.

·  Consider white matter tracts and tractography-based atlases, e.g.,

Meola A, Yeh FC, Fellows-Mayle W, et al. (2016). Human connectome-based tractographic atlas of the brainstem connections and surgical approaches. Neurosurgery;79(3):437-55.

·  Consider not only electrode location but also the electrical distribution of the electrode, e.g.

Haegelen C, Baumgarten C, Houvenaghel J-F, Zhao Y, Pearon J, Drapier S, et al. (2018) Functional atlases for analysis of motor and neuropsychological outcomes after medial globus pallidus and subthalamic stimulation. PLoS ONE 13(7):

7. Discuss various use and functionality of brain atlases in major stereotactic surgery workstations (see some examples are in your [18]).

8. Lines 391-2 “As a logical deduction, it is also reasonable to envisage, for the near future, a scenario in which these data will be integrated and available on the Web”.

Such a solution was provided to the stereotactic and functional neurosurgery community two decades ago free of charge along with the way of generation and display of the probabilistic functional atlas - no one was willing to deposit any data.

Nowinski WL, Belov D, Benabid AL: A community-centric Internet portal for stereotactic and functional neurosurgery with a probabilistic functional atlas. Stereotactic and Functional Neurosurgery 2002;79:1-12.

The minor issues are the following:

Lines 147-8: “Schaltenbrand and Bailey published a brain atlas, probably the world’s most widely used one”.

I would argue. Please justify this statement.

Line 200: “All horizontal sections are parallel to the intercommissural plane”.

This statement is not correct as these sections are slanted by about 7 degrees.

Line 325: “www.cerefy.com

This site is not accessible anymore.

Author Response

Response to Reviewer #2

Reviewer: This paper nicely presents a brief history of stereotactic atlases. Its major limitation is that it focuses predominantly on early developments and is not complete. This work can benefit from the following extensions and enhancements.

Response: We express our gratitude for the reviewer's favorable assessment of our manuscript and his/her guidance for manuscript implementation. We have taken the reviewer's comments into account and have extensively revised our study, addressing the points that were previously overlooked. As a result, the revised manuscript is substantially different from the previous version. We sincerely hope that our diligent efforts will be recognized and that the new manuscript will be considered a significant improvement.

Reviewer Q1: Provide a section on early electronic atlases, in particular the pioneering work done by Bertran et al Keely et al.

Bertrand G, Olivier A, Thompson CJ. (1974). Computer display of stereotaxic brain maps and probe tracts. Acta Neurochir Suppl 21:235-243.

Kall BA, Kelly PJ, Goerss S, Frieder G. (1985). Methodology and clinical experience with Computed Tomography and a computer-resident stereotactic atlas. Neurosurgery;17(3):400-407.

Response: We added a section on early electronic atlases.

9.1 Early Computer-based Atlases

Probably, the first computer program with digitized (and scalable) stereotactic atlases was developed by Bertrand et al. (Bertrand G, Olivier A, Thompson CJ. Computer display of stereotaxic brain maps and probe tracts. Acta Neurochir (Wien). 1974;Suppl 21:235-43. doi: 10.1007/978-3-7091-8355-7_30. PMID: 4412564.). The authors transformed map line drawings into digital data, stored in the computer memory, by tracing enlarged photographs of the transparent line drawings of the Atlas with the mechanical pen of an analogue X Y plotter. The drawings obtained could then be displayed on the viewing screen of a Tektronix type 4002 "Computer Graphics Terminal". The software could be interrogated using a number of questions, the answers to which were readily available from simple measurements on the stereotaxic ventriculogram: the distance between the commissures, the height of the thalamus above the intercommissural plane, if visible; the width of the third ventricle. Answers to these questions were simply typed on the terminal keyboard. They are used to compress or expand the corresponding dimensions of the map to match the individual patient's brain, at least as far as the X-ray landmarks are concerned.

A digital version of the Schaltenbrand and Wahren atlas resident in a computer was created by Kall et al. (Kall BA, Kelly PJ, Goerss S, Frieder G. Methodology and clinical experience with computed tomography and a computer-resident stereotactic atlas. Neurosurgery. 1985 Sep;17(3):400-7. doi: 10.1227/00006123-198509000-00002. PMID: 3900793.). The authors, for the first time, were able to digitalize the atlas and to warp it onto CT scans of individual patients using polar coordinates. The CT data base were reformatted in planes having the same orientation as horizontal microscopic sections of a computer-based stereotactic atlas. The software was able to aligns the atlas to fit within computer-based anatomical boundaries by polar scaling. Thus, the reformatted CT slice becomes a labeled atlas of the individual patient’s brain.

Reviewer Q2.  Discuss in more detail the AC and PC

In particular:

  • The Talairach AC point landmark is beyond the AC structure, and it is missing in the Talairach 1988 atlas at the coronal CA=0 plate i.e. at the center of the coordinate system (in my version of the atlas Prof. Talairach added this landmark manually).
  • Although the Talairach and Schaltenbrand atlas are based on the AC and PC, these landmarks are defined differently.
  • On MRI patients’ scans used for surgery planning the AC and PC are usually defined as center points.
  • Various definitions and AC and PC landmarks and the resulting differences are addressed e.g., in 

Nowinski WL: Modified Talairach landmarks. Acta Neurochirurgica 2001;143(10);1045-1057.

 Response: We added a section addressing this point:

7.2 The Talairach-Nowinski System

The Talairach proportional grid system transformation is a useful normalization method in clinical practice due to its simplicity and low computational cost. However, the original atlas has certain limitations. Some Talairach landmarks are not present on the plates, and the locations of others are contradictory to their definitions. The AC and PC landmarks are positioned outside their respective structures. Although the AC point is considered the origin of the Talairach coordinate system, the coronal atlas plate passing through the origin (marked as CA=0) does not contain the AC landmark. This inconsistency in the atlas can lead to errors in landmark identification and grid placement. The L landmark on the axial plates is located a few millimeters away from the grid, while the P landmark lies beyond the Talairach grid. Additionally, the R landmark is absent altogether. The Talairach-Tournoux atlas does not cover the entire Talairach space, resulting in the unavailability of certain landmarks on different plates. For example, the S and I landmarks are missing on the axial plates, the A and P landmarks are not present on the coronal plates, and the L and R landmarks are absent on the sagittal plates. Moreover, the intercommissural plate is also missing from the atlas.

The fact that the Talairach landmarks and Talairach grid are not consistent impacts the accuracy of atlas-to-data registration. In order to overcome the above problems, Nowinski (Nowinski WL: Modified Talairach landmarks. Acta Neurochirurgica 2001;143(10);1045-1057) introduced a new, equivalent set of landmarks called the Talairach-Nowinski landmarks (or, in brief, T-N landmarks). The introduced definitions of the T-N landmarks allowed for narrowing the searching space in the brain when identifying the cortical landmarks automatically. The cortical landmarks can be calculated approximately on three planes: intercommissural plane, coronal plane passing through the AC, and coronal plane passing through the PC.  To define the intercommissural plane, different intercommissural lines were introduced to be defined on MRI: 1. The central intercommissural line is passing through the centres of the anterior commissure and the posterior commissure on the midsagittal (interhemispheric) plane. 2. The tangential intercommissural line is tangential dorso-posteriorly to the anterior commissure and ventroanteriorily to the posterior commissure on the midsagittal plane.  3. The internal intercommissural line as a distance between the internal intercommissural landmarks (Figure 6)

Figure 6. To define the intercommissural plane, different intercommissural lines are introduced to be defined on MRI. h): original intercommissural distance; hc): central intercommissural distance between the central intercommissural landmarks; hi): internal intercommissural distance between the internal intercommissural landmarks; ht): tangential intercommissural distance between the tangential intercommissural landmarks.

The internal intercommissural distance is the closest to the original Talairach intercommissural distance. The relative intercommissural error is only 0.5%, as opposed to the central and tangential intercommissural distances resulting in high relative intercommissural errors, about 10%. On the other hand, the internal and central intercommissural distances result in a high maximum displacement error at the cortex amounting to about 11 mm while the tangential intercommissural distance gives only 1 mm error. The sensitivity of the internal intercommissural distance to the actual location of the intercommissural plane is high amounting to above 10%. On the other hand, the sensitivity of the central intercommissural distance is very low, below 0.5%.

Thus, using the T-N system, it is possible to find cortical landmarks easily using the intercommisural plane, but a selection depending on the application and user control over the placement of the intercommisural plane is necessary. For stereotactic and functional neurosurgery, the internal intercommissural distance is the most suitable to provide a high accuracy for subcortical structures. In localisation analysis in human brain mapping research, a high accuracy has to be achieved at the cortex and the tangential intercommissural line is superior.

Reviewer Q3. It may be worth mentioning about automated methods for the calculation of the Talairach landmarks and atlas-to-scan mapping, e.g.

Nowinski WL, Qian G, Bhanu Prakash KN, Hu Q, Aziz A: Fast Talairach Transformation for magnetic resonance neuroimages. Journal of Computer Assisted Tomography 2006;30(4):629-41.

 Response: We added a section addressing this point:

9.2 Atlas-to-scan Warping

While the Talairach atlas remains the most commonly used system for reporting coordinates in neuroimaging studies, the absence of an actual 3D image of the original brain used in its construction has severely limited the ability of researchers to automatically map locations in 3D anatomical images (Lacadie CM, Fulbright RK, Rajeevan N, Constable RT, Papademetris X. More accurate Talairach coordinates for neuroimaging using non-linear registration. Neuroimage. 2008 Aug 15;42(2):717-25. doi: 10.1016/j.neuroimage.2008.04.240. Epub 2008 Apr 30. PMID: 18572418; PMCID: PMC2603575.).

Warping atlases to pre-operative patient data has been one solution to solve the issue of lack of three-dimensionality of traditional printed atlases and to obtain individual functional data directly on a patient 3D-MRI. Registration of the atlas to the patient is typically achieved using linear scaling techniques based primarily on the length of the AC–PC line and the width of the third ventricle.  Reliable atlas-to-brain warping, however, requires to set multiple parametersand complex softwares packages developed to this purpose. They differ not only in functionality provided, but also in brain normalization methods applied and brain atlas, being the Talairach and Tournoux atlas and/or the MNI template are typically employed. The two most common methods currently used for warping were: (1) a direct piecewise-linear scaling of each individual subject into Talairach space such as that which is applied in, for example, the AFNI software package (Cox RW. AFNI: software for analysis and visualization of functional magnetic resonance neuroimages. Comput Biomed Res. 1996 Jun;29(3):162-73. doi: 10.1006/cbmr.1996.0014. PMID: 8812068.) or (2) mapping each individual subject into a common reference space (the most common reference space is the Montreal Neurological Institute (MNI) space) (A. C. Evans, D. L. Collins, S. R. Mills, E. D. Brown, R. L. Kelly and T. M. Peters, "3D statistical neuroanatomical models from 305 MRI volumes," 1993 IEEE Conference Record Nuclear Science Symposium and Medical Imaging Conference, San Francisco, CA, USA, 1993, pp. 1813-1817 vol.3, doi: 10.1109/NSSMIC.1993.373602.) and then applying a piecewise-linear conversion for mapping MNI coordinates to Talairach coordinates.

Nowinski et al. (Nowinski WL, Fang A, Nguyen BT, Raphel JK, Jagannathan L, Raghavan R, Bryan RN, Miller GA. Multiple brain atlas database and atlas-based neuroimaging system. Comput Aided Surg. 1997;2(1):42-66. doi: 10.1002/(SICI)1097-0150) have developed a digital atlas that incorporated data from three print atlases including Ono et al., 1990, (M. Ono, S. Kubik, C.D. Abernathy Atlas of the Cerebral Sulci. Georg Thieme Verlag/Thieme Medical Publishers, Stuttgart, Germany, 1990), Schaltenbrand and Wahren, 1977, Talairach and Tournoux, 1988.  In order to register the atlas to a subject or patient, a piece-wise linear approach is used to transform the atlas to the MR volume. 

The Talairach transformation (TT) is at present the most widespread method for brain normalization and atlas to data warping (Nowinski WL, Qian G, Bhanu Prakash KN, Hu Q, Aziz A. Fast Talairach Transformation for magnetic resonance neuroimages. J Comput Assist Tomogr. 2006 Jul-Aug;30(4):629-41. doi: 10.1097/00004728-200607000-00013. PMID: 16845295.). Conceptually it is simple from an anatomy and transformation standpoints. The TT is based on the 8 Talairach point landmarks: anterior commissure (AC) and posterior commissure (PC) located on the midsagittal plane (MSP) and 6 cortical landmarks determining the extents of the brain in the anterior (A), posterior (P), left (L), right (R), superior (S), and inferior (I) directions. The transformation warps the source volume image into the target image piecewise linearly with 13 degrees of freedom (DOFs).

The described methods have the limit of accounting only for the overall size and orientation of the brain , but not for any other variable. It has been shown that the overall shape of the MNI template is taller, longer and has larger temporal lobes than the Talairach brain. Brett et al. (Brett, M., Christoff, K., Cusack, R., & Lancaster, J. (2001). Using the Talairach atlas with the MNI template. Neuroimage, 13(6), 85-85.) created the mni2tal transform which maps MNI space into Talairach space using two linear transformation matrices, but notes that the technique is merely an approximation. To improve on the above-mentioned techniques, researchers have studied the disparity between Talairach coordinates derived from different methods and have applied various affine transforms to optimize the coordinates between MNI space and Talairach space (Carmack PS, Spence J, Gunst RF, Schucany WR, Woodward WA, Haley RW. Improved agreement between Talairach and MNI coordinate spaces in deep brain regions. Neuroimage. 2004 May;22(1):367-71. doi: 10.1016/j.neuroimage.2004.01.022. PMID: 15110028.; Chau W, McIntosh AR. The Talairach coordinate of a point in the MNI space: how to interpret it. Neuroimage. 2005 Apr 1;25(2):408-16. doi: 10.1016/j.neuroimage.2004.12.007. PMID: 15784419.; Lancaster, J. L. (1997). The Talairach Daemon, a database server for Talairach atlas labels. Neuroimage, 5, S633.). While affine transformations contain more information than the piecewise-linear transformations, they still do not account for differences in brain shape. Non-linear registrations employ thousands of parameters instead of the 12 parameters used typically for affine registrations and therefore can account for these regional shape differences. Theoretically, nonlinear and high degree-of-freedom (DOF) brain warping methods are more powerful and accurate than piecewise linear methods with low/medium DOFs. However, accuracy is not the only critical factor enabling a wide use of brain warping methods. Other factors include cost (time of execution), user acceptance, ease of use, validation, and availability

In order to achieve a fully automatic and almost realtime warping of a brain atlas against neuroimages, Nowinski et al. (Nowinski WL, Qian G, Bhanu Prakash KN, Hu Q, Aziz A. Fast Talairach Transformation for magnetic resonance neuroimages. J Comput Assist Tomogr. 2006 Jul-Aug;30(4):629-41. doi: 10.1097/00004728-200607000-00013. PMID: 16845295) introduced a Fast Talairach Transformation (FTT).  The FTT calculates the landmarks and warps the Talairach and Tournoux atlas fully automatically in about 5 sec on a standard computer. After overlaying the individualized atlas on the patient- or subject-specific brain scan, this scan can be interactively labeled, segmented, and globally searched for a given structure by means of the atlas. The FTT was validated for 215 MR scans demonstrating localization accuracy of about 1 voxel for the AC and. PC, and half a voxel for the cortical landmarks.

Atlases digitalization offers a series of advantages. Indeed, it allows for rapid planning, suitable for surgical procedures, increases the accuracy of targeting by using the multiple. orientations, multiple atlases, atlases in contour representation superimposed on the data, global and local registrations with any clearly visible landmarks, real-time interactive atlas warping feasible any time, allows the neurosurgeon to plan more sophisticated trajectories by displaying the trajectory on all three planes and in 3-D, lowers the invasiveness and risk of the surgical procedure by reducing the number of microelectrodes necessary fr exploration (Nowinski, Wieslaw L. (2001). Computerized Brain Atlases for Surgery of Movement Disorders. Seminars in Neurosurgery, 12(2), 183–194. doi:10.1055/s-2001-17125).

Reviewer Q4. Elaborate more about properties, especially limitations, of the Talairach and Tournoux 1988 and the Schaltenbrand and Wahren atlases, e.g.:

Niemann K, Naujokat C, Pohl G, Wollner C, von Keyserling KD (1994) Verification of the Schaltenbrand and Wahren stereotactic atlas. Acta Neurochir (Wien) 129(1–2):72–81. 

Niemann K, van Nieuwenhofen I (1999) One atlas-three anatomies: relationships of the Schaltenbrand and Wahren microscopic data. Acta Neurochir (Wien) 141:1025–1103. 

Nowinski WL, Thirunavuukarasuu A: Quantification of spatial consistency in the Talairach and Tournoux stereotactic atlas. Acta Neurochirurgica 2009 151(10):1207-13.

Nowinski WL, Liu J, Thirunavuukarasuu A: Quantification and visualization of three-dimensional inconsistency of the ventrointermediate nucleus of the thalamus in the Schaltenbrand-Wahren brain atlas. Acta Neurochirurgica 2008;150(7):647-53.

 Response: We added a section addressing this point:

7.1 Consistencies and inconsistencies of the Talairach and Tournoux 1988 and the Schaltenbrand and Wahren atlases

Because of methodology of their construction, traditional printed atlases have potential limitations in terms of 3D inconsistencies and spatial sparseness. For instance, the Schaltenbrand axial plates were not acquired exactly in the intercommissural plane but are rotated 7 degrees clockwise (Nowinski, Wieslaw L. (2001). Computerized Brain Atlases for Surgery of Movement Disorders. Seminars in Neurosurgery, 12(2), 183–194. doi:10.1055/s-2001-17125). Also, the atlases are constructed based on a few brains only: the Talairach and Tournoux atlas on a single brain and the SW atlas microseries on two different brains (three various hemispheres) despite using 111 brains as the initial material. Niemann and Nieuwenhofen (Niemann K, van Nieuwenhofen I. One atlas - three anatomies: relationships of the Schaltenbrand and Wahren microscopic data. Acta Neurochir (Wien). 1999;141(10):1025-38. doi: 10.1007/s007010050479. PMID: 10550646.) analyzed 3-D position of 21 anatomical structures in the three series of the Schaltenbrand and Wahren atlas, after digitally interpolation and volumetric representation. 3D-rendering showed that sagittally the thalamus is 10% larger than the frontally represented and 40% larger than the horizontally sectioned thalamus.  Thus, in order to match it to the sagittal series, the frontal series has to be widened in lateral direction by 19%, in anteroposterior and dorsobasal (vertical) direction it has to be compressed by 5 and 9%, respectively.  In contrast, the distance of structures from the midline in the horizontal and sagittal series is very similar. The horizontal series is, however, 25% smaller than the sagittal one in anteroposterior and 17% in vertical direction. On average, thalamic nuclei in the right hemisphere of brain LXXVIII (horizontal microscopic series). Spatial overlap between corresponding thalamic nuclei from the three series amounted to only 0±28% when superimposed in the AC-PC reference space.

Nowinski and Thirunavuukarasuu (Nowinski WL, Liu J, Thirunavuukarasuu A. Quantification and visualization of three-dimensional inconsistency of the ventrointermediate nucleus of the thalamus in the Schaltenbrand-Wahren brain atlas. Acta Neurochir (Wien). 2008 Jul;150(7):647-53; discussion 653. doi: 10.1007/s00701-007-1419-3. Epub 2008 Jun 18. PMID: 18560749) analyzed 3D inconsistencies of the ventrointermediate nucleus (VIM) of the thalamus in the Schaltenbrand-Wahren atlas.  The 3D models were reconstructed from the axial, coronal and sagittal microseries, respectively, by applying a shape-based method. All 3D models, placed in the SW coordinate system, were compared quantitatively in terms of location (centroids), size (volumes), shape (normalised eigen values), orientation (eigen vectors), and mutual spatial relationships (overlaps and inclusions).  A significant 3D inaccuracy within each orientation was found, confirming the findings of Niemann and Nieuwenhofen (CITARE).

As the Talairach-Tournoux print atlas was obtained from a single brain specimen, consistency in the three projections could be expected. Nevertheless, process of atlas construction by cutting the specimen sagittally and interpolating the other two orientations manually resulted in spatial inconsistency across the orthogonal orientations. Nowinski and Thirunavuukarasuu (Nowinski WL, Thirunavuukarasuu A. Quantification of spatial consistency in the Talairach and Tournoux stereotactic atlas. Acta Neurochir (Wien). 2009 Oct;151(10):1207-13. doi: 10.1007/s00701-009-0364-8. PMID: 19730778.) examined consistency problem by analyzing the complete atlas simultaneously on all three orthogonal planes. Two measures were introduced: consistency and discrepancy. The consistency refers to uniformity of labeling across the orthogonal orientations and is calculated at the grid points, being the points of intersections of all three atlas planes: axial, coronal and sagittal. The discrepancy determines the spatial offset in labeling across orientations caused by manual interpolation of the original print atlas. According to. The authors measurements, the Talairach-Tournoux atlas has 27.4% consistency and 37.7% inconsistency, being the thalamus the most consistent structure (85.7% consistency, 5.4% inconsistency).

Accordingly, uncritical plotting of recordings into the bidimensional atlases coordinate spaces would jeopardize the consistency of the morphological and electrophysiological databases. Also, there is the danger that electrophysiological findings may be traced to different subnuclei of the thalamus dependent on the chosen atlas series and the different rigid or elastic matching procedures applied (Niemann K, Naujokat C, Pohl G, Wollner C, von Keyserlingk D. Verification of the Schaltenbrand and Wahren stereotactic atlas. Acta Neurochir (Wien). 1994;129(1-2):72-81. doi: 10.1007/BF01400876. PMID: 7998500). Understanding of this problem paved the way for more coherent atlases interpolation and construction of consistent 3D atlases.

Reviewer Q5. Discuss in more detail the differences between the Schaltenbrand and Bailey atlas (claimed by the authors to be “probably the world’s most widely used one”) and the Schaltenbrand and Wahren atlas (to be the most prevalent in surgical workstations).

Response: We omitted the sentence  “probably the world’s most widely used one”. We added the following note:

  1. The “Talairach” and “Schaltenbrand” Atlases of the CT era

The Schaltenbrand-Wahren brain atlas was published in 1977 [20]. It differs from the Schaltenbrand and Bailey atlas being based on myelin-stained sections instead of histological sections.  The atlas contains 100 instead of 97 sections and it is overall organized in a different manner, with different levels of detail and different areas covered.

Reviewer Q6. The way of the atlas use is vaguely addressed.

In particular:

  • The atlas can be used preoperatively, intraoperatively, and postoperatively, for instance

Nowinski WL: Computerized brain atlases for surgery of movement disorders. Seminars in Neurosurgery 2001;12(2):183-194.

Response: We added the following note in the Introduction:

Preoperatively, the stereotactic atlas is an important reference guide to avoid targeting errors due to insufficient anatomical details and, in general, to provide basic orientation during functional neurosurgery procedures [1]. Intraoperatively, it can also function as a navigation tool and provide information about the neuroanatomy surrounding the target area, the structures along the stereotactic path, and the spatial relationships between the electrode tip and critical structures such as the optic tract. (Nowinski WL, Thirunavuukarasuu A, Benabid AL. The cerefy clinical brain atlas: enhanced edition with surgical planning and intraoperative support. New York: Thieme; 2005). Additionally, it can aid in data storage. Postoperatively, the atlas can be used to evaluate the spatial positioning of deep brain stimulation (DBS) or lesion placement (Nowinski WL, Yang GL, Yeo TT. Computer-aided stereotactic functional neurosurgery enhanced by the use of the multiple brain atlas database. IEEE Trans Med Imaging 2000;19(1):62-9).

Also, we added a section on clinical applications of brain atlases:

Clinical Applications

Human brain atlases have been widely used in clinical applications, particularly in stereotactic and functional neurosurgery. Initially, digital atlases such as The Electronic Clinical Brain Atlas (Nowinski, W.L.; Bryan, R.N.; Raghavan, R. The Electronic Clinical Brain Atlas. Multiplanar Navigation of the Human Brain; Thieme: New York, NY, USA, 1997) were used offline in the operating room, and later incorporated into surgical workstations like the StealthStation to assist neurosurgeons in pre-, intra-, and post-operative support. Brain atlases are useful for planning target and trajectory, identifying structures intersected by the trajectory, and specifying the structures already traversed by the electrode. They also measure distances to important structures, provide neuroanatomic and vascular context, and examine the precision of electrode placement. While current targeting techniques focus on the target point, atlas reporting cerebrovascular anatomy may assist in better placement of the entry point. Indeed, a small change in the entry point of a DBS electrode, occurring quite commonly especially in frameless DBS procedures, can lead to a major change in the trajectory. Nowinski et al. (Nowinski WL, Chua BC, Volkau I, Puspitasari F, Marchenko Y, Runge VM, Knopp MV: Simulation and assessment of cerebrovascular damage in deep brain stimulation using a stereotactic atlas of vasculature and structure derived from multiple 3T and 7T scans. Journal of Neurosurgery 2010;113(6):1234-41) created an atlas of structure and vasculature with > 900 vessels, with the smallest being 90 μm in diameter (figure 13), it can be used to analyze the track-brain spatial relationship allowing the DBS electrode to be placed more effectively, potentially reducing the invasiveness of the DBS procedure.

Figure 13. A 3-dimensional rendering of arterial and venous structures.  3D-atlas of human vasculature can be used to analyze in DBS to analyse track-brain spatial relationship allowing the DBS electrode to be placed more effectively (from www.nowinbrain.org).

Brain atlases are also employed in neuroradiology, where they can assist in neuroimage interpretation by segmenting and labeling brain scans including pathological reporting, facilitating processing of multi-detector scans, and enabling communication between physicians and patients. Brain atlases also have potential in stroke management, neurology, and psychiatry (Nowinski, W.L.; Qian, G.; Bhanu Prakash, K.N.; Thirunavuukarasuu, A.; Hu, Q.M.; Ivanov, N.; Parimal, A.S.; Runge, V.M.; Beauchamp, N.J. Analysis of ischemic stroke MR images by means of brain atlases of anatomy and blood supply territories. Acad. Radiol. 2006, 13, 1025–1034; Nowinski, W.L. Human brain atlases in stroke management. Neuroinformatics 2020, 18, 549–567; Nowinski, W.L.; Chua, B.C. Bridging neuroanatomy, neuroradiology and neurology: Three-dimensional interactive atlas of neurological disorders. Neuroradiol. J. 2013, 26, 252–262). For example, they can provide automated processes for predicting, diagnosing, and treating strokes, demonstrate various locations of brain damage and resulting neurologic deficits, and generate neuroanatomic volumes of interest for statistical analysis of psychiatric disorders like schizophrenia (Sim, K.; Yang, G.L.; Loh, D.; Poon, L.Y.; Sitoh, Y.Y.; Verma, S.; Keefe, R.; Collinson, S.; Chong, S.A.; Heckers, S.; et al. White matter abnormalities and neurocognitive deficits associated with the passivity phenomenon in schizophrenia: A diffusion tensor imaging study. Psychiatry Res. 2009, 172, 121–127).

Reviewer Q6.1 ·  This survey addresses targeting, but the atlas can be used for entry point localization, for instance

Nowinski WL, Chua BC, Volkau I, Puspitasari F, Marchenko Y, Runge VM, Knopp MV: Simulation and assessment of cerebrovascular damage in deep brain stimulation using a stereotactic atlas of vasculature and structure derived from multiple 3T and 7T scans. Journal of Neurosurgery 2010;113(6):1234-41

Response: We added the following note in the section 12, entitled “Brain Atlas Applications”

While current targeting techniques focus on the target point, atlas reporting cerebrovascular anatomy may assist in better placement of the entry point. Indeed, a small change in the entry point of a DBS electrode, occurring quite commonly especially in frameless DBS procedures, can lead to a major change in the trajectory. Nowinski et al. (Nowinski WL, Chua BC, Volkau I, Puspitasari F, Marchenko Y, Runge VM, Knopp MV: Simulation and assessment of cerebrovascular damage in deep brain stimulation using a stereotactic atlas of vasculature and structure derived from multiple 3T and 7T scans. Journal of Neurosurgery 2010;113(6):1234-41) created an atlas of structure and vasculature with > 900 vessels, with the smallest being 90 μm in diameter (figure 13), it can be used to analyze the track-brain spatial relationship allowing the DBS electrode to be placed more effectively, potentially reducing the invasiveness of the DBS procedure.

Figure 13. A 3-dimensional rendering of arterial and venous structures.  3D-atlas of human vasculature can be used to analyze in DBS to analyse track-brain spatial relationship allowing the DBS electrode to be placed more effectively (from www.nowinbrain.org).

Reviewer Q6.2 ·  Address the development of electrophysiology databases, e.g.,

Finnis K, Starreveld Y, Parrent A, Sadikot A, Peters T. (2003). Three-dimensional database of subcortical electrophysiology for image-guided stereotactic functional neurosurgery. IEEE Trans Med Imag.;22(1):93–104.

Response: We discussed this point in the section 11, entitled “Probabilistic and functional atlas”

A different atlasing method consists in building an atlas from functional data collected in a population of subjects or patients [32]. The data can be either preoperative electrophysiological recordings or clinical exploration data (e.g., points that provoke arm dyskinesias or somesthetic perceptions on the hand) or postoperative tuning data (e.g., a contact that provokes heat sensation or diplopia).

To enhance the precision of surgical targeting, various authors have gathered and analyzed functional data from subcortical structures of multiple individuals during stereotaxy (C. J. Thompson, T. L. Hardy, and G. Bertrand, “A system for anatomical. and functional mapping of the human thalamus,” Comput. Biomed. Res., vol. 10, pp. 9–24, 1977; R. R. Tasker, L. W. Organ, and P. A. Hawrylyshyn, The Thalamus and Midbrain of Man. Springfield, IL: Charles C. Thomas, 1982; C. Giorgi, U. Cerchiari, G. Broggi, P. Birk, and A. Struppeler, “Digital. image processing to handle neuroanatomical information and neurophysiological data,” Appl. Neurophysiol., vol. 48, pp. 30–33, 1985; M. Yoshida, K. Okada, A. Nagase, S. Kuga, M. Shirahama, M.  Watanabe, and S. Kuramoto, “Neurophysiological atlas of the human. thalamus and adjacent structures. Computer-assisted mapping,” Appl. Neurophysiol., vol. 45, pp. 406–409, 1982.). Data were transformed in alphanumeric codes that were used to standardize the electro-physiological data, which were then normalized to an anatomical atlas to develop composite functional maps. By registering electroanatomic observations from several patients to a common coordinate space, it became possible to examine functional organization in relation to anatomic structures. Bertrand et al. (G. Bertrand, J. Blundell, and R. Musella, Electrical Stimulation of the Internal Capsule and Neighboring Structures During Stereotactic Procedures. Philadelphia, PA: Harvey Cushing Soc., Apr. 20, 1963.) first introduced this technique, displaying a rough somatotopic organization of corticobulbar and corticospinal fibers in the posterior limb of the internal capsule, derived from a group of 26 patients normalized to a representative plate of the Schaltenbrand Bailey atlas. Later, they expanded this technique to include interactive recording and display of physiologic responses collected during surgery (C. J. Thompson, T. L. Hardy, and G. Bertrand, “A system for anatomical and functional mapping of the human thalamus,” Comput. Biomed. Res., vol. 10, pp. 9–24, 1977.). Tasker et al. (R. R. Tasker, L. W. Organ, and P. A. Hawrylyshyn, The Thalamus and. Midbrain of Man. Springfield, IL: Charles C. Thomas, 1982.) published the results of microstimulation in 1982, which was conducted on 9383 sites during 198 procedures, primarily for Parkinson's disease and chronic pain. Tasker's work still represents one the most comprehensive analysis of electrophysiological observations obtained through microstimulation available to date. Finnis et al. (Finnis K, Starreveld Y, Parrent A, Sadikot A, Peters T. (2003). Three-dimensional database of subcortical electrophysiology for image-guided stereotactic functional neurosurgery. IEEE Trans Med Imag.;22(1):93–104) nonlinearly registered electrophysiological data obtained from 88 patients (106 procedures) via microelectrode recording and electrical stimulation to patient’s MRI, and then to a high-resolution MRI reference brain. Once registered, the authors found the clustering of interpatient physiologic responses within the thalamus, globus pallidus, subthalamic nucleus, and adjacent structures. These data were in turn be registered to a three-dimensional patient MRI within an image-guided visualization program enabling prior to surgery the delineation of surgical targets, anatomy with high probability of containing specific cell types, and functional borders. Advantages of this method included the use of nonlinear registration to accommodate for interpatient anatomical variability and the avoidance of digitized versions of printed atlases of anatomy as a common database coordinate system.

Reviewer Q6.3 ·  Consider white matter tracts and tractography-based atlases, e.g., 

Meola A, Yeh FC, Fellows-Mayle W, et al. (2016). Human connectome-based tractographic atlas of the brainstem connections and surgical approaches. Neurosurgery;79(3):437-55.

Response: We discussed this point in the section 13, entitled “Ongoing Projects”

Furthermore, the combination of anatomic information together with electrophysiology, cytoarchitectural, connectomics and molecular findings provides a unique knowledge of the human brain as a whole. (Toga AW, Thompson PM, Mori S, Amunts K, Zilles K. Towards multimodal atlases of the human brain. Nat Rev Neurosci. 2006 Dec;7(12):952-66. doi: 10.1038/nrn2012. PMID: 17115077; PMCID: PMC3113553.). In recent years, there has been a significant increase in human brain-related projects, initiatives, and programs. These efforts, which are often large, advanced, government-led, and/or well-funded, include The Human Connectome Project (Van Essen, D.C.; Smith, S.M.; Barch, D.M.; Behrens, T.E.J.; Yacoub, E.; Ugurbil, K. The WU-Minn Human Connectome Project: An overview. NeuroImage 2013, 80, 62–79.), The Allen Brain Atlas (Sunkin, S.M.; Ng, L.; Lau, C.; Dolbeare, T.; Gilbert, T.L.; Thompson, C.L.; Hawrylycz, M.; Dang, C. Allen Brain Atlas: An integrated spatio-temporal portal for exploring the central nervous system. Nucleic Acids Res. 2013, 41, D996–D1008)., The Big Brain (Amunts, K.; Lepage, C.; Borgeat, L.; Mohlberg, H.; Dickscheid, T.; Rousseau, M.É.; Bludau, S.; Bazin, P.L.; Lewis, L.B.; Oros-Peusquens, A.M.; et al. Bigbrain: An ultrahigh-resolution 3D human brain model. Science2013340, 1472–14759), The CONNECT project. (Assaf, Y.; Alexander, D.C.; Jones, D.K.; Bizzi, A.; Behrens, T.E.; Clark, C.A.; Cohen, Y.; Dyrby, T.B.; Huppi, P.S.; Knoesche, T.R.; et al. The CONNECT project: Combining macro- and micro-structure. Neuroimage 2013, 80, 273–282.), the Brainnetome project (Jiang, T. Brainnetome: A new -ome to understand the brain and its disorders. Neuroimage 201380, 263–272.), The BRAIN Initiative. (Jorgenson, L.A.; Newsome, W.T.; Anderson, D.J.; Bargmann, C.I.; Brown, E.N.; Deisseroth, K.; Donoghue, J.P.; Hudson, K.L.; Ling, G.S.; MacLeish, P.R.; et al. The BRAINInitiative: Developing technology to catalyse neuroscience discovery. Philos. Trans. R. Soc. B Biol. Sci. 2015, 370, 20140164), The Blue Brain Project (Markram, H.; Muller, E.; Ramaswamy, S.; Reimann, M.W.; Abdellah, M.; Sanchez, C.A.; Ailamaki, A.; Alonso-Nanclares, L.; Antille, N.; Arsever, S.; et al. Reconstruction and simulation of neocortical microcircuitry. Cell 2015, 163, 456–492.), The Human Brain Project (Amunts, K.; Ebell, C.; Muller, J.; Telefont, M.; Knoll, A.; Lippert, T. The Human Brain Project: Creating a European research infrastructure to decode the human brain. Neuron 2016, 92, 574–581.), the Chinese Color Nest Project. (Zuo, X.N.; He, Y.; Betzel, R.F.; Colcombe, S.; Sporns, O.; Milham, M.P. Human connectomics across the life span. Trends Cogn. Sci. 201721, 32–45. ), the Japanese Brain/MINDS project (Sadato, N.; Morita, K.; Kasai, K.; Fukushi, T.; Nakamura, K.; Nakazawa, E.; Okano, H.; Okabe, S. Neuroethical issues of the Brain/MINDS Project of Japan. Neuron 2019, 101, 385–389), and SYNAPSE (Synchrotron for Neuroscience—an Asia-Pacific Strategic Enterprise) (Chin, A.L.; Yang, S.M.; Chen, H.H.; Li, M.T.; Lee, T.T.; Chen, Y.J.; Lee, T.K.; Petibois, C.; Cai, X.; Low, C.M.; et al. A synchrotron X-ray imaging strategy to map large animal brains. Chin. J. Phys. 2020, 65, 24–32.) These projects aim to map the structural and functional connections of the brain, understand brain circuits and behavior, develop technology to advance neuroscience discovery, simulate neocortical micro-circuitry, and create brain-inspired information technology.

Reviewer Q6.4 ·  Consider not only electrode location but also the electrical distribution of the electrode, e.g.

Haegelen C, Baumgarten C, Houvenaghel J-F, Zhao Y, Pearon J, Drapier S, et al. (2018) Functional atlases for analysis of motor and neuropsychological outcomes after medial globus pallidus and subthalamic stimulation. PLoS ONE 13(7):

Response: We discussed this point in the section 11, entitled “Probabilistic and functional atlas”

A further step in the creation of functional atlases, was proposed by Haegelen et al. (Haegelen C, Baumgarten C, Houvenaghel J-F, Zhao Y, Péron J, Drapier S, et al. (2018) Functional atlases for analysis of motor and neuropsychological outcomes after medial globus pallidus and subthalamic stimulation. PLoS ONE 13(7): e0200262. https://doi.org/10.1371/journal.pone.0200262) collecting functional data not only of the electrode location, but also the electrical distribution of the current, represented by the volume theoretically activated by each stimulation. In order to model the electric field, the authors used a pre-defined 3D Gaussian. Based on the lead’s characteristics, a monopolar stimulation and on the quasi-static potential equation, the stimulation influence covered approximately a 3 mm-radius sphere around each stimulation contact. As a common space, authors chose a multi-subject MR template created from a population of patients with Parkinson’s Disease, named the ParkMedAtlis template.

Reviewer Q7. Discuss various use and functionality of brain atlases in major stereotactic surgery workstations (see some examples are in your [18]).

Response: We discussed this point in section 10, entitled: “3D-rendered Atlases from Histological Sections and Multiatlas Collections”

Neurosurgical workstations commonly feature computerized brain atlases, such as the Cerefy Electronic Brain Atlas Library and/or Cerefy Brain Atlas Geometrical Models. These atlases can be found in various systems, including the StealthStation (Medtronic Surgical Navigation Technologies), Target and iPlan (BrainLAB AG), SurgiPlan (Elekta Instrument), SNN 3 Image-Guided Surgery System (Surgical Navigation Specialists), and the neurosurgical robot NeuroMate (Integrated Surgical Systems). Additionally, the Cerefy brain atlas libraries are being evaluated by companies such as Prosurgics, Renishaw, Cedara Software, and Z-KAT. Other companies, such as Tyco/Radionics and Stryker/Leibinger, have also developed their own digital versions of the SWand TT print atlases. The COMPASS System of Stereotactic Medical Systems and the CASS system of MIDCO also offer electronic atlases.

Reviewer Q8. Lines 391-2 “As a logical deduction, it is also reasonable to envisage, for the near future, a scenario in which these data will be integrated and available on the Web”.

Such a solution was provided to the stereotactic and functional neurosurgery community two decades ago free of charge along with the way of generation and display of the probabilistic functional atlas - no one was willing to deposit any data.

Nowinski WL, Belov D, Benabid AL: A community-centric Internet portal for stereotactic and functional neurosurgery with a probabilistic functional atlas. Stereotactic and Functional Neurosurgery 2002;79:1-12.

Response: We re-analyzed the issue and discussed this point in section 11, entitled “Probabilistic and functional atlas”

An Internet portal for stereotactic and functional neurosurgery combined with a PFA was also developed in which any neurosurgeon was able to query about best targets as well as input his or her own microrecordings, convert them into probabilistic functional maps, and merge them if needed with the maps of other neurosurgeon (Nowinski WL, Belov D, Benabid AL. A community-centric internet portal for stereotactic and functional neurosurgery with a probabilistic functional atlas. Stereotact Funct Neurosurg. 2002;79(1):1-12. doi: 10.1159/000069499. PMID: 12677100). Thus, the PFA overcomes the limitations of existing computerized brain atlases whereas the public availability enabled to construct a progressively more accurate PFA of the human brain for standard and future stereotactic targets by the neurosurgical and neuroscience communities.  Although this portal is no longer in use, sharing of the PFA through community-based portal and the progressive implementation by neurosurgeons represent a paradigm shift from manufacturer centric to community centric approach to stereotactic and functional neurosurgery.

Reviewer 3 Report

Comments and Suggestions for Authors

In the era of the development of deep brain stimulation (DBS) as one of the most promising therapeutic surgical techniques, a critical review of the development of stereotactic atlases and the assessment of their value is most welcome. Such a task was undertaken by the authors of the reviewed work, specifying the goal as follows: "The aim of this text is to review their principal characteristics highlighting the milestones of their evolution."... . It so happens that this task has now been undertaken by other authors, presenting a more advanced literature review than currently assessed, so the question arises, what is the superiority of the current work over other published ones?

The Introduction chapter contains a well-known description of the development of stereotaxy and the principles of its application. Few publications are listed (out of a total of 38 in the bibliography, which is short for a review article), which, in addition to the names of renowned atlases and their authors, would critically list the pros and cons of the most popular ones today. Please don't take this the wrong way, but the level of information transfer is comparable to Wikipedia. Experts on the subject know what the intercommissural line is and how it is carried out, and oriented in relation to the thalamus, which is presented in the following subsections. They know the value of Schaltenbrand and Bailey atlases and atlases showing variations of human diencephalon by Van Buren and Borke, “Talairach” and “Schaltenbrand” Atlases of the CT era etc. Figure captions contain misspellings that are confused with numbers. Next, the authors present Digital histological Atlases and Three-dimensional Atlases, which do not bring new content (even critical content) to those presented so far. One of the most interesting parts of the work is the approach adopted by the authors group [24]. Further content, apart from a cursory description of 3D-rendered Atlases from Histological Sections and Probabilistic Atlases, does not bring anything new to the existing knowledge about stereotaxy. There are no specific Conclusions at the end of the Discussions which in fact, does not exist, only short, general Conclusions.

References

The list of references does not include some most important review articles regarding the most contemporary modern human brain atlases related to stereotaxy. Although authors recognize and appreciate the works of Nowinski they omitted his latest important and valuable review article Nowinski WL. Evolution of Human Brain Atlases in Terms of Content, Applications, Functionality, and Availability. Neuroinformatics. 2021;19(1):1-22.. The work of Toga AW, Thompson PM, Mori S, Amunts K, Zilles K. Towards multimodal atlases of the human brain. Nat Rev Neurosci. 2006;7(12):952-966. is also of great interest but was not mentioned by the authors.  

It is not difficult to be critical of the editorial preparation of References, where one can find at least four ways of citing (often incomplete) textbooks, atlases, and original works, nothing like the MDPI way of citation.

Comments on the Quality of English Language

In the era of the development of deep brain stimulation (DBS) as one of the most promising therapeutic surgical techniques, a critical review of the development of stereotactic atlases and the assessment of their value is most welcome. Such a task was undertaken by the authors of the reviewed work, specifying the goal as follows: "The aim of this text is to review their principal characteristics highlighting the milestones of their evolution."... . It so happens that this task has now been undertaken by other authors, presenting a more advanced literature review than currently assessed, so the question arises, what is the superiority of the current work over other published ones?

The Introduction chapter contains a well-known description of the development of stereotaxy and the principles of its application. Few publications are listed (out of a total of 38 in the bibliography, which is short for a review article), which, in addition to the names of renowned atlases and their authors, would critically list the pros and cons of the most popular ones today. Please don't take this the wrong way, but the level of information transfer is comparable to Wikipedia. Experts on the subject know what the intercommissural line is and how it is carried out, and oriented in relation to the thalamus, which is presented in the following subsections. They know the value of Schaltenbrand and Bailey atlases and atlases showing variations of human diencephalon by Van Buren and Borke, “Talairach” and “Schaltenbrand” Atlases of the CT era etc. Figure captions contain misspellings that are confused with numbers. Next, the authors present Digital histological Atlases and Three-dimensional Atlases, which do not bring new content (even critical content) to those presented so far. One of the most interesting parts of the work is the approach adopted by the authors group [24]. Further content, apart from a cursory description of 3D-rendered Atlases from Histological Sections and Probabilistic Atlases, does not bring anything new to the existing knowledge about stereotaxy. There are no specific Conclusions at the end of the Discussions which in fact, does not exist, only short, general Conclusions.

References

The list of references does not include some most important review articles regarding the most contemporary modern human brain atlases related to stereotaxy. Although authors recognize and appreciate the works of Nowinski they omitted his latest important and valuable review article Nowinski WL. Evolution of Human Brain Atlases in Terms of Content, Applications, Functionality, and Availability. Neuroinformatics. 2021;19(1):1-22.. The work of Toga AW, Thompson PM, Mori S, Amunts K, Zilles K. Towards multimodal atlases of the human brain. Nat Rev Neurosci. 2006;7(12):952-966. is also of great interest but was not mentioned by the authors.  

It is not difficult to be critical of the editorial preparation of References, where one can find at least four ways of citing (often incomplete) textbooks, atlases, and original works, nothing like the MDPI way of citation.

Author Response

Response to Reviewer #3

Reviewer Comments: In the era of the development of deep brain stimulation (DBS) as one of the most promising therapeutic surgical techniques, a critical review of the development of stereotactic atlases and the assessment of their value is most welcome. Such a task was undertaken by the authors of the reviewed work, specifying the goal as follows: "The aim of this text is to review their principal characteristics highlighting the milestones of their evolution."... . It so happens that this task has now been undertaken by other authors, presenting a more advanced literature review than currently assessed, so the question arises, what is the superiority of the current work over other published ones?

The Introduction chapter contains a well-known description of the development of stereotaxy and the principles of its application. Few publications are listed (out of a total of 38 in the bibliography, which is short for a review article), which, in addition to the names of renowned atlases and their authors, would critically list the pros and cons of the most popular ones today. Please don't take this the wrong way, but the level of information transfer is comparable to Wikipedia. Experts on the subject know what the intercommissural line is and how it is carried out, and oriented in relation to the thalamus, which is presented in the following subsections. They know the value of Schaltenbrand and Bailey atlases and atlases showing variations of human diencephalon by Van Buren and Borke, “Talairach” and “Schaltenbrand” Atlases of the CT era etc. Figure captions contain misspellings that are confused with numbers. Next, the authors present Digital histological Atlases and Three-dimensional Atlases, which do not bring new content (even critical content) to those presented so far. One of the most interesting parts of the work is the approach adopted by the authors group [24]. Further content, apart from a cursory description of 3D-rendered Atlases from Histological Sections and Probabilistic Atlases, does not bring anything new to the existing knowledge about stereotaxy. There are no specific Conclusions at the end of the Discussions which in fact, does not exist, only short, general Conclusions.

References

The list of references does not include some most important review articles regarding the most contemporary modern human brain atlases related to stereotaxy. Although authors recognize and appreciate the works of Nowinski they omitted his latest important and valuable review article Nowinski WL. Evolution of Human Brain Atlases in Terms of Content, Applications, Functionality, and Availability. Neuroinformatics. 2021;19(1):1-22.. The work of Toga AW, Thompson PM, Mori S, Amunts K, Zilles K. Towards multimodal atlases of the human brain. Nat Rev Neurosci. 2006;7(12):952-966. is also of great interest but was not mentioned by the authors.  

It is not difficult to be critical of the editorial preparation of References, where one can find at least four ways of citing (often incomplete) textbooks, atlases, and original works, nothing like the MDPI way of citation.

Response: We express our gratitude for the reviewer's review and assessment of our manuscript. We have taken the severe reviewer's comments into account and have extensively revised our study, addressing many points that were previously overlooked. Indeed, our manuscript was intended as “brief historical overview” of atlasing. We recognize that a similar approach is out of date and did not represent a substantial contribution to the current literature. For this, we have made a thorough revision of our manuscript extending the discussion to other major aspects concerning the use, limits, and innovation in brain atlasing. Relevant literature was reviewed, analyzed and it is now reported in this new manuscript version including several new sections.  As a result, the revised manuscript is substantially a new submission. Although there are still other similar review articles in the literature, we believe that our attempt to provide an historical overview together with a perspective on modern use of brain atlas, can be publishable in a special issue dedicated to DBS and functional neurosurgery.   

We sincerely hope that the Reviewer would offer another possibility to our manuscript and efforts.

We addressed minor points also, including misspelling in figure captions and reference format.

Suggested papers were reviewed and included in the reference list. This, together with further 60 references reviewed and discussed.

Conclusions were revised as well. Even though they are still concise, all major issues were discussed in the main text of the study.

Round 2

Reviewer 2 Report

Comments and Suggestions for Authors

Line 127 ref [18] - please check if it is correct here

Reviewer 3 Report

Comments and Suggestions for Authors

The Authors addressed most of my remarks and queries, which influenced the higher scientific values and informative content of the paper.

They introduced the suggested and new references and discussed their content in comparison to the proposed new techniques including the inventions by the authors. The authors' work also significantly increased the editorial value of the manuscript.

I will agree with the statement to accept the paper in its current form if the authors include the graphical content of Figures 12 and 13 which are lacking. Perhaps it is due to the .docs to .pdf conversion and the graphics are still present in the Word file which was not currently provided for the review. If it happens, I accept the paper for publication and leave the last correction for the correspondence between the Authors and the Brain Sci Editorial Office.

Comments on the Quality of English Language

Minor English spelling for the final discussion with the Type-setter.